# Dynamic Diffusion Schrödinger Bridge in Astrophysical Observational Inversions

**Ye Zhu**[*1,2], **Duo Xu**[3], **Zhiwei Deng**[4], **Jonathan C. Tan**[5,6], **Olga Russakovsky**[1]

[1]Department of Computer Science, Princeton Univeristy, Princeton NJ, USA
[2]LIX, École Polytechnique, IP Paris, Palaiseau, France
[3] Canadian Institute for Theoretical Astrophysics (CITA), University of Toronto, Toronto, Canada
[4] Google DeepMind, Mountain View, CA, USA
[5] Department of Astronomy, University of Virginia, Charlottesville VA, USA
[6] Department of Space, Earth & Environment, Chalmers University of Technology, Gothenburg, Sweden
ye.zhu@polytechnique.edu,xuduo@cita.utoronto.ca,
zhiweideng@google.com,jct6e@virginia.edu,olgarus@princeton.edu

## Abstract

We study Diffusion Schrödinger Bridge (DSB) models in the context of dynamical astrophysical systems, specifically tackling observational inverse prediction tasks within Giant Molecular Clouds (GMCs) for star formation. We introduce the *Astro-DSB* model, a variant of DSB with the pairwise domain assumption tailored for astrophysical dynamics. By investigating its learning process and prediction performance in both physically simulated data and in real observations (the Taurus B213 data), we present two main takeaways. First, from the astrophysical perspective, our proposed paired DSB method improves interpretability, learning efficiency, and prediction performance over conventional astrostatistical and other machine learning methods. Second, from the generative modeling perspective, probabilistic generative modeling reveals improvements over discriminative pixel-to-pixel modeling in Out-Of-Distribution (OOD) testing cases of physical simulations with *unseen initial conditions and different dominant physical processes*. Our study expands research into diffusion models beyond the traditional visual synthesis application and provides evidence of the models' learning abilities beyond pure data statistics, paving a path for future physics-aware generative models which can align dynamics between machine learning and real (astro)physical systems. [2]

## 1   Introduction

Diffusion models (DMs) (Sohl-Dickstein et al., 2015; Ho et al., 2020; Song et al., 2020), sometimes also known as score-based generation models (SGMs), have become a popular probabilistic modeling approach in machine learning for many data generation applications (Dhariwal and Nichol, 2021; Ho et al., 2022; Rombach et al., 2022). A recent line of work improves the vanilla design by reformulating the mapping trajectory learning in the context of the Schrödinger bridge problem (Schrödinger, 1932), which allows for imposing domain-specific restrictions on the learned trajectory, leading to Diffusion Schrödinger Bridge models (DSB) (Wang et al., 2021; De Bortoli et al., 2021; Shi et al., 2023). These DSB models are theoretically grounded and have proven effective in vision-related applications such as image generation and translation (Liu et al., 2023; Gushchin et al., 2024). However, their modeling capabilities and limits could extend beyond vision, offering a new opportunity to test these generative

---

[*]Work mainly completed when YZ was a postdoc at Princeton University.
[2]Code available at https://github.com/L-YeZhu/AstroDSB

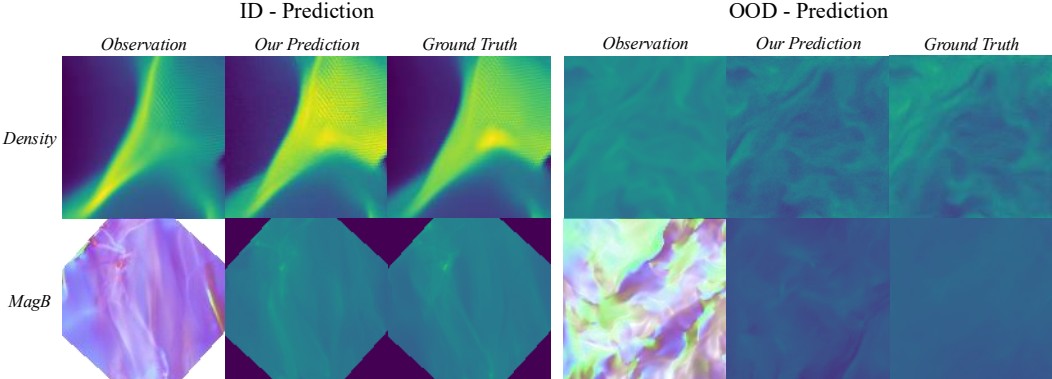

Figure 1: **We introduce *Astro-DSB* for astrophysical inverse predictions from observables within Giant Molecular Clouds (GMCs).** We visualize the observables, predicted results, and ground truth physical distributions on the density (top) and magnetic field (bottom) for in-distribution (*ID*) and out-of-distribution (*OOD*) testing cases, with the OOD defined via changes in initial physical conditions and dominant dynamical processes in the simulations.

formulations in the context of real-world physical systems, where data are governed by complex dynamics and observational constraints rather than pixel statistics.

In this work, we study a real-world physical problem, focusing on a key class of inverse prediction tasks in astrophysics, particularly within the context of star formation (McKee and Ostriker, 2007; Larson, 2003; Madau and Dickinson, 2014). Specifically, we aim to better understand and predict the *physical states* within Giant Molecular Clouds (GMCs) (Shu et al., 1987), which is a major component of the interstellar medium (ISM) (Spitzer Jr, 2008) that directly regulates star formation rates. The goal of our task is to infer true physical distributions, such as density or magnetic fields, from partial observables, as illustrated in Fig. 1.

As a critical and active research area in astrophysics, multiple methods have been developed over the past decades to tackle this challenge. Given that observational inverse prediction problems are inherently intractable, early approaches often relied on *astrostatistical methods* such as power-law fitting (Maschberger and Kroupa, 2009), which combine theoretical simplifications with observational statistical priors (detailed in Sec. 2.2 and Sec. 4.2). However, these methods typically suffer from large prediction errors and limited generalization on out-of-distribution (OOD) testing cases under shifted physical initial conditions, due to often oversimplified physical assumptions and restricted capacity for modeling complex non-linearities.

Given the increasing capability of neural networks in modeling large-scale data with highly complicated nonlinearities, the astrophysical community has recently started to explore machine learning approaches (Yang et al., 2024; Xu et al., 2020a) to overcome previous limitations. Among multiple ML architectures, conditional diffusion generative models (DGMs) (Sohl-Dickstein et al., 2015; Ho et al., 2020; Song et al., 2020) stand out as recent state-of-the-art (SOTA) frameworks with large improvements in prediction accuracy over conventional methods for several critical physical quantities, such as GMCs density distribution (Xu et al., 2023b) and magnetic field strengths (Xu et al., 2025). These methods typically take the observables as input conditions and output the predicted physical distribution maps through the distribution mapping ability of DGMs trained on simulation data. However, despite the enhanced prediction accuracy, such conditional diffusion models share a few constraints, including a physics-ML misalignment due to the Gaussian prior from the original DM formulation (also detailed in Sec. 3.2), as well as the known expensive computational cost for learning DGMs.

To this end, we introduce *Astro-DSB*, a variant of Diffusion Schrödinger Bridge model tailored to astrophysics with better physics-ML state alignment and improved learning efficiency. Specifically, compared to existing DGMs-based methods that learn a mapping between the Gaussian prior and the true physical distribution with the observable as the condition, *Astro-DSB* removes the reliance on the Gaussian prior from the DGMs formulation and learn the direct mapping between observables and ground-truth physical states. This design also helps to *accelerate the learning efficiency by 75%*

compared to the previous DGM-based SOTA. More concretely, *Astro-DSB* leverages the intrinsic deterministic mapping property between the observation and ground truth physical state and adopts a paired assumption between two boundary distributions [3]. Within this high-level Schrödinger bridge ML framework, we further integrate the observables as an enhanced condition to facilitate the bridge learning process. Intuitively, this can be mathematically considered as learning a *conditional Doob's h-transform*, thus differentiating from the vanilla paired DSB models in computer vision tasks (Liu et al., 2023) and scientific applications such as protein designs (Didi et al., 2023).

Our comprehensive experiments on both molecular density and magnetic field tasks validate the effectiveness of our proposed method. In addition to the comparable or superior SOTA performance in ID testing cases, we also examine the performance under distribution shifts using both physically simulated data and **real astrophysical observations from Taurus B213**. In contrast to conventional vision tasks, where OOD is often heuristically defined, astrophysical systems provide scientifically meaningful definitions of distributional shifts through variations in initial conditions and dominant physical processes (Wang and Abel, 2009; Wu et al., 2015). These configurations reflect realistic physical diversity, making them especially suitable for testing distribution-level generalization. Through this evaluation, we uncover the practical boundaries of probabilistic modeling and its advantages in scientific domains such as astrophysics.

The rest of the paper is organized as follows: we present related works in Sec. 2; Sec. 3 details the core technical designs; Sec. 4 explains our controlled experiments and comprehensive analysis comparing against conventional astrophysical and existing ML approaches; and we conclude by discussing the limitations and future directions in Sec. 5. To summarize our main contributions, our work offers two key takeaway messages: for scientific applications such as astrophysical inverse prediction, the proposed *Astro-DSB* framework offers an efficient and accurate alternative to conventional modeling approaches; for the broader generative modeling community, our study highlights the advantage of distribution-level learning in enabling meaningful generalization under physically grounded distribution shifts, extending the application scenarios from simulation data to real-world observations from Taurus B213 (Li and Goldsmith, 2012; Palmeirim et al., 2013).

## 2 Related work

### 2.1 Schrödinger bridge problem and generative modeling

The machine learning method we leverage in this work is closely related to one line of active research efforts in the Schrödinger bridge problem (SB) with application to generative models. Specifically, given the similar underlying problem of learning the mapping between two arbitrary probabilistic distributions, prior works (Wang et al., 2021; De Bortoli et al., 2021; Shi et al., 2023, 2022; Somnath et al., 2023; Liu et al., 2023) have explored improving generative modeling by formulating the boundary distribution matching problem within the Optimal Transport (OT) map of SB (Cuturi, 2013). Specifically, Wang et al. (2021) proposes to learn a generative model via entropy interpolation with a Schrödinger bridge in a two-stage manner. While this work relies on a *static* SB problem, later works (De Bortoli et al., 2021; Shi et al., 2023; Liu et al., 2023; Gushchin et al., 2024; Ksenofontov and Korotin, 2025) explicitly propose to compute the *dynamic* SB with the additional time moments $t \in [0, 1]$ via the score-based diffusion models to learn the desired optimal path, under different specific restrictions (e.g., continuous v.s. discrete data space and unpaired v.s. paired domain translation). These prior works largely target static domains in practical applications, leaving open the question of how DSB performs in settings where the dynamic trajectory is physically meaningful, which is a question we directly investigate in this work.

### 2.2 Inverse astrostatistical predictions for star formation

The real-world physical problem we address in this paper focuses on a key class of inverse prediction tasks in astrostatistics, particularly within the context of star formation (McKee and Ostriker, 2007; Larson, 2003; Madau and Dickinson, 2014). We focus on the *dynamic physical processes* within Giant Molecular Clouds (GMCs) (Shu et al., 1987), a major component of the interstellar medium

---

[3]Terminology note: The term *"paired DSB"* is used here in the ML sense, which refers to the sample-level conditioning between observables and physical states, rather than in the classical unpaired Schrödinger Bridge formulation based on probabilistic coupling.

(ISM) (Spitzer Jr, 2008) that directly regulates star formation rates. The goal of our task is to infer true physical distributions, such as density or magnetic fields, from partial observables. In contrast to image generation or translation tasks where input-output mappings are static and visually structured, our problem setup reflects dynamic and physically driven distributional transitions.

Traditionally in astronomy, such observational inverse prediction problems have been commonly approached via *astrostatistical methods*, which often combine theoretical simplifications with statistical priors (also in Sec. 4). For instance, molecular density is commonly estimated using two- or three-component power-law conversion derived under idealized physical assumptions about emission properties and cloud structure (Bisbas et al., 2021, 2023). While effective within limited regimes, these methods tend to lack generalization ability when applied to systems governed by varying physical conditions and dynamics, making their performance unreliable in unseen testing cases.

Grounding DSB modeling within this context thus enables a more meaningful evaluation of probabilistic models beyond synthetic benchmarks (Liu et al., 2023; Gushchin et al., 2024; Ksenofontov and Korotin, 2025; Su et al., 2023). Notably, we leverage a key strength of the astrophysical setting: the ability to define *Out-of-Distribution (OOD)* conditions in a principled and physically grounded way through simulations with varying initial conditions and dominant physical processes. In addition to standard evaluation on *In-Domain* (ID) cases, our experiments explicitly assess OOD generalization, offering new insights into the robustness of probabilistic modeling in scientific prediction tasks.

# 3 Diffusion Schrödinger bridge for astrophysical observational inversion

## 3.1 Problem statement and preliminary in astronomy and ML

**GMCs observational prediction.** Studying the precise physical distributions within Giant Molecular Clouds (GMCs) is challenging yet essential for understanding the star formation process in astronomy. This challenge mainly arises from the highly complex and nonlinear physical dynamics within the forward evolution and the observational constraints.

Given a physical initial condition $\mathbf{x}_0$, we assume that the GMCs evolve under a set of governing physical laws as modeled in previous astrophysical studies (Wu et al., 2015, 2020; Hsu et al., 2023). This results in an equilibrium physical state $\mathbf{x}_1$, which we denote as $\mathbf{x}_1 = \mathcal{H}(x_0)$, where $\mathcal{H}$ is a highly complex, nonlinear function incorporating self-gravity, magnetic fields, radiative heating, and cooling processes. While the forward process $\mathcal{H}$ is typically well-defined in controlled simulations, direct access to $\mathbf{x}_1$ is *rarely available in practice*. Instead, we often only observe a limited projection $\mathbf{y}$ of the true physical state. Therefore, our objective is to infer the full distribution $\mathbf{x}_1$ from partial observable $\mathbf{y}$, featuring a class of observational inverse prediction problems that lie at the core of many tasks in astrostatistical modeling. We further assume an observational model in the form of:

$$\mathbf{y} = \mathcal{F}(\mathbf{x}_1) + \varepsilon = \mathcal{F}(\mathcal{H}(\mathbf{x_0})) + \varepsilon, \tag{1}$$

where $\mathcal{F}$ represents the observational process (e.g., line-of-sight integration, instrumental response) and $\varepsilon$ denotes corresponding observational noise in the measurement. Given this formulation, our core task becomes a conditional probabilistic inference problem, where the goal is to estimate the posterior distribution $p(\mathbf{x}_1|\mathbf{y})$.

**Preliminary on dynamic DSB.** Consider two distributions $p_0, p_1$ and a reference path measure $\mathbb{Q} \in \mathcal{P}(\mathcal{C})$, the dynamic SB problem seeks to find a path measure $\mathbb{P}^{SB} \in \mathcal{P}(\mathcal{C})$ such that:

$$\mathbb{P}^{SB} = \mathrm{argmin}_{\mathbb{P}}\{\mathrm{KL}(\mathbb{P}|\mathbb{Q}) \; : \; \mathbb{P}_0 = p_0, \; \mathbb{P}_1 = p_1\}, \tag{2}$$

with $\mathcal{C} = \mathrm{C}([0,1], \mathbb{R}^d)$ defined as the space of continuous functions from time space $[0, 1]$ to $\mathbb{R}^d$. SB can be formalized as an entropy-regularized OT problem describing the following forward and backward SDEs:

$$
\begin{aligned}
d\mathbf{x}_t &= [f(\mathbf{x}, t) + g(t)^2 \nabla \log \Psi(\mathbf{x}, t)]dt + g(t)dw, \\
d\mathbf{x}_t &= [f(\mathbf{x}, t) - g(t)^2 \nabla \log \hat{\Psi}(\mathbf{x}, t)]dt + g(t)d\overline{w}.
\end{aligned}
\tag{3}
$$

Here, $f(\cdot, t)$ is the linear drift, $g(t)$ controls the diffusion intensity over time, and $dw$ represents the standard Wiener processes for modeling stochastic noise. The wave functions $\Psi$ and $\hat{\Psi}$ are

time-varying energy potentials that solve the following coupled PDEs:

$$\frac{\partial \Psi(\mathbf{x}, t)}{\partial t} = -\nabla \Psi(\mathbf{x}, t)^T f(\mathbf{x}, t) - \frac{1}{2} g(t)^2 \Delta \Psi(\mathbf{x}, t),$$

$$\frac{\partial \hat{\Psi}(\mathbf{x}, t)}{\partial t} = \nabla \cdot (\hat{\Psi}(\mathbf{x}, t) f(\mathbf{x}, t) + \frac{1}{2} g(t)^2 \Delta \hat{\Psi}(\mathbf{x}, t),$$

(4)

with $\Psi(\mathbf{x}, 0)\hat{\Psi}(\mathbf{x}, 0) = p_0(\mathbf{x})$ and $\Psi(\mathbf{x}, 1)\hat{\Psi}(\mathbf{x}, 1) = p_1(\mathbf{x})$. Recent works have shown that score-based diffusion models provide a practical numerical solver for the DSB problem by parameterizing the time-dependent scores associated with forward and backward SDEs (De Bortoli et al., 2021; Shi et al., 2023; Liu et al., 2023). Specifically, instead of explicitly solving the Schrödinger system via wave potentials $\Psi$ and $\hat{\Psi}$, SGMs learn the forward and reverse score functions $\nabla \log p_t(\mathbf{x})$ through denoising score matching, simulating the optimal stochastic process by integrating the learned SDEs. This formulation enables the DSB problem to be learned from data via stochastic sampling and bypasses the need for explicit access to density functions $p_t(\mathbf{x})$, which are often intractable in real-world scientific applications. In our work, we build on this framework and introduce a pairwise-constrained variant of the DSB model tailored to astrophysical observational inversion tasks, as described in the following section.

## 3.2 Diffusion Schrödinger bridge for astrophysical observational prediction (Astro-DSB)

**Core technical motivation and relation to prior works.** Our core technical motivation seeks to better align the states $\mathbf{x}_t$ in score-based diffusion models with the dynamic evolution processes governing GMCs, which leads to two key differences from existing literature.

*Within this field of observational prediction in GMCs*, existing state-of-the-art (SOTA) ML-based methods (Xu et al., 2023a,b, 2025) leverage conditional diffusion denoising probabilistic models (DDPMs), framing the observational data $\mathbf{y}$ as the condition. While achieving superior performance compared to traditional astrostatistical methods, conditional DDPMs *introduce unnecessary Gaussian prior assumptions*, which are not naturally aligned with the physics of GMCs. Our proposed Schrödinger bridge framework removes this assumption by directly modeling the transition between observable $\mathbf{y}$ and the true physical distribution $\mathbf{x}_1$, resulting in a distribution-level probabilistic mapping *without reliance on artificial prior structures*, thus offering better ML interpretability compred to conditional DDPMs.

*From the physics-aligned modeling perspective*, Li et al. (2025) recently propose another variant of diffusion bridge with a two-stage learning process (data-driven pretraining and physics-aligned finetuning) for physical field reconstruction and super-resolution. In contrast, our approach adopts a single-stage learning framework to learn the transition path directly without pretraining, and systematically evaluates both in-distribution performance and *OOD generalization* to assess robustness, further extending the prediction to real observational data.

**M0: Pairwise matching in marginal distributions.** Similar to many other physical tasks (Li et al., 2025), our observational prediction problem exhibits a pairwise connection between marginal distributions $p_0$ and $p_1$ established by $\mathbf{y}$ and $\mathbf{x}_1$, respectively. Based on this, we follow the prior work $I^2SB$ (Liu et al., 2023) for a special case of the SB problem. Specifically, $I^2SB$ presents a tractable class of SB that eliminates the couplings of the wave functions from Eq. 3, and propose an analytical form for the implementation of the posterior distribution:

$$p(\mathbf{x}_t|\mathbf{y}, \mathbf{x}_1) = \mathcal{N}(\mathbf{x}_t; \frac{\bar{\sigma}_t^2}{\bar{\sigma}_t^2 + \sigma_t^2}\mathbf{y} + \frac{\sigma_t^2}{\bar{\sigma}_t^2 + \sigma_t^2}\mathbf{x}_1, \frac{\bar{\sigma}_t^2 \sigma_t^2}{\bar{\sigma}_t^2 + \sigma_t^2}\mathbf{I}),$$

(5)

where $\sigma_t^2 := \int_0^t \beta_\tau d\tau$ and $\bar{\sigma}_t^2 := \int_t^1 \beta_\tau d\tau$ denote accumulated variances. This forms the base design of our proposed *Astro-DSB*. However, we observe that this baseline framework alone is insufficient for accurately predicting true physical distributions from constrained observational inputs.

**M1: Noise perturbation alignment between ML and observables.** Compared to conventional image translation (Liu et al., 2023) and other physical tasks such as field reconstruction (Li et al., 2025) that take a clean data input $\mathbf{y}$, we note that explicitly modeling the noise perturbation term $\epsilon$ in observables $\mathbf{y}$ in Eq. 5 offers a simple yet critical alignment to physical assumptions in observational

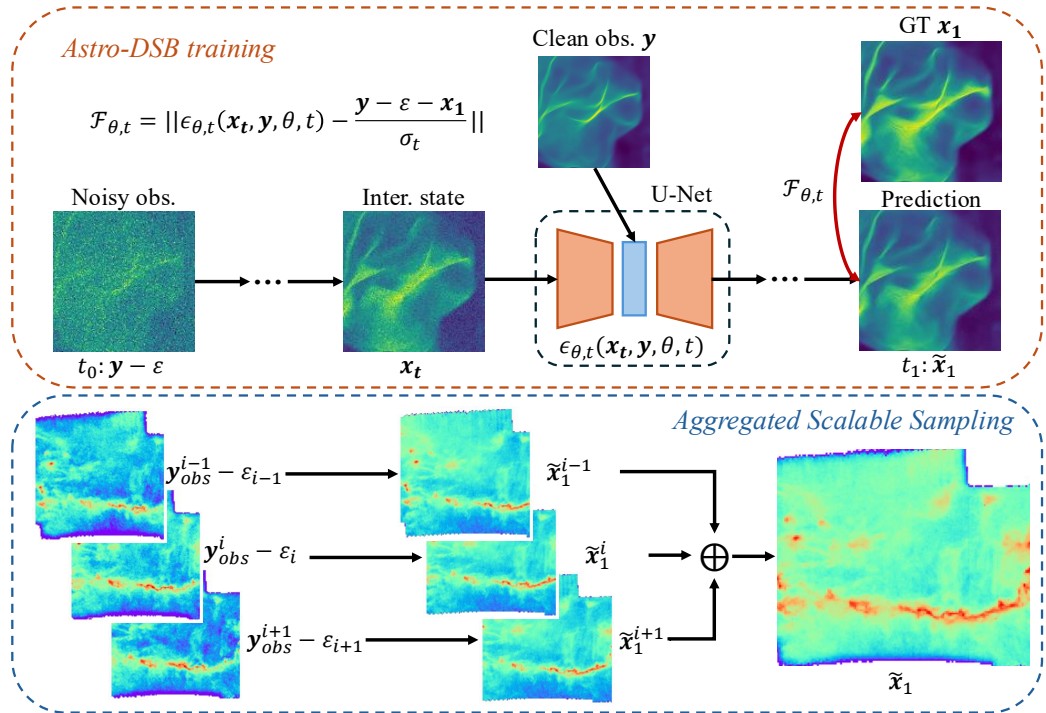

Figure 2: **Illustration pipeline of our proposed *Astro-DSB* method.** In the training stage, we propose to learn the DSB model under the pairwise matching assumption with observational noises alignment $\varepsilon$ and enhancement observables $\mathbf{y}$. In inference, we adopt an aggregated scalable sampling strategy to infer physical distributions from larger observables, such as Taurus B213.

prediction scenarios described in Eq. 1. In this work, we assume that the noise introduced from the observation process follows a weighted Gaussian, denoted as $\epsilon \sim \lambda \mathcal{N}(0, I)$. Following statistical conventions in astrophysical studies, we set $\lambda = 0.1$ throughout the experiments. This modification ensures that our *Astro-DSB* framework respects the uncertainty structure of constrained observables, refining the targeted conditional posterior distribution to be $p(\mathbf{x}_1 | \mathbf{y} - \varepsilon)$.

**M2: Observable enhancement within DSB model.** Compared to the vanilla $I^2$SB framework, we further propose to enhance the learning process by incorporating the clean observable input $\mathbf{y}$ as an additional intermediate condition at each time step $t$. This modification aims to facilitate the learning of the score function $\epsilon_{\theta,t}$, implemented through U-Net architecture (Ronneberger et al., 2015), by consistently "reminding" the model of the observational constraint throughout the trajectory. Empirically, we find this design critical for achieving robust prediction performance within the DSB framework and helps simultaneously obtain lower loss values with a faster convergence speed.

**Patch-based training and aggregated scalable sampling.** Following the methodological designs described above, we train the proposed *Astro-DSB* using the reparameterized objective function:

$$\mathcal{F}_{\theta,t} = ||\epsilon_{\theta,t}(\mathbf{x}_t, \mathbf{y}, t; \theta) - \frac{\mathbf{y} - \varepsilon - \mathbf{x}_1}{\sigma_t}||^2, \tag{6}$$

where $\mathbf{x}_t$ takes the analytical form from Eq. 5. Given that astrophysical data often present large spatial resolutions, we adopt a patch-based training strategy that is widely used in both vision and scientific applications (Li et al., 2025; Yang et al., 2019), with cropped patches from the original simulation data. Specifically, $\mathbf{x}_1$ and $\mathbf{y}$ are cropped into smaller patches of consistent size to facilitate efficient learning while preserving local physical structures.

For inference, in contrast to the padding-based techniques used in prior works (Li et al., 2025), we design an aggregated sampling strategy to restore higher-resolution prediction results. Given a large testing observable $\mathbf{y}_{obs}$ (e.g., real-world survey data), we partition $\mathbf{y}_{obs}$ into overlapping patches

$\mathbf{y}_{obs}^i$ that match the training resolution. We then independently run inference on each patch to obtain corresponding predictions $\tilde{\mathbf{x}}_1^i$, and aggregate across all patches to reconstruct the final predicted distribution $\tilde{\mathbf{x}}_1$. Fig. 2 illustrates the described pipeline, with detailed algorithms in the appendices.

# 4 Experiments

## 4.1 Astrophysical data

**Magnetohydrodynamics simulations.** Similar to previous work (Xu et al., 2023b, 2025), our training data are drawn from synthetic magnetohydrodynamics (MHD) simulations. Specifically, we follow the simulation setup from Wu et al. (2020) and Hsu et al. (2023), which are conducted using the MUSCL-Dedner method and HLLD Riemann solver in the adaptive mesh refinement (AMR) code Enzo (Dedner et al., 2002; Wang and Abel, 2009; Bryan et al., 2014). These simulations model key physical processes including self-gravity, magnetic fields, radiative heating, and cooling, based on a photodissociation model (Wu et al., 2015). We consider the *molecular density* ($\rho$, measured in the number of H nuclei $n_H/cm^{-3}$) and magnetic field strength ($\mathcal{B}$, measured in $\mu G$) as the primary quantities for prediction in this work. For molecular density, we collected data from 7179 simulation trials, split into 5707 training and 1472 testing cases following an 8:2 ratio. For magnetic field strength, we obtained 19100 training and 6370 testing cases. All data are pre-processed by a logarithmic transformation followed by normalization to [0,1], and resized to $128 \times 128$ patches.

Additionally, we generate 1680 independent tests with different dominate physical process assumptions as the OOD testing cases to evaluate generalization. Specifically, unlike the ID MHD simulations code ENZO (Bryan et al., 2014) initialized with colliding and non-colliding molecular clouds, OOD data are produced using a different MHD code, ORION2 (Li et al., 2021), which excludes gravity but includes driven turbulence, and employs a different set of initial conditions. While the fundamental MHD equations are similar between two simulation codes, the physical processes and initial setups differ significantly, thus clearly distinguishing the ID and OOD cases under physically meaning shifts.

**Synthetic and real observables.** After generating the simulation data, we define the observables $\mathbf{y}$ following conventions in astrophysical studies. For molecular density, we adopt the projected mass surface density maps (also known as the "column density", measured in $N_H/cm^{-2}$) as the observables $\mathbf{y}_\rho$. For the magnetic field, the synthetic observable $\mathbf{y}_B$ consists of the column density, the orientation angles of the dust continuum polarization vector, and line-of-sight (LOS) nonthermal velocity dispersion. In addition to these synthetic observables, we also evaluate the proposed *Astro-DSB* on real observational data from the Taurus B213 region, using column density maps provided by Palmeirim et al. (2013). B213 is one of closest star-forming filamentary structures located in the Taurus molecular cloud (Li and Goldsmith, 2012). Unlike the synthetic data where ground-truth distributions are available, the Taurus B213 observation $\mathbf{y}_{B213}$ lacks a precise reference for physical states, and is instead constrained by indirect measurements from complementary observations. This real-data evaluation demonstrates the applicability of *Astro-DSB* in practical scenarios where ground truth is unavailable, and highlights its potential for scientific discovery in observational settings.

## 4.2 Comparing methods and evaluations

**Astrostatistical methods.** The conventional methods involve astrostatistical techniques to infer physical distributions with simplified physical assumptions. For molecular density prediction, existing popular techniques include statistical conversions such as 2-component (2PLF) and 3-component power-law fitting (3PLF). For magnetic field, Davis–Chandrasekhar–Fermi (DCF) method (Davis, 1951; Chandrasekhar and Fermi, 1953; Beck, 2016) has been the classic method for estimating the magnetic field strength, which relies on the assumption of an equipartition between the magnetic field energy and the turbulent kinetic energy of the gas.

**Machine learning methods.** Recent studies have also adopted machine learning techniques, which can be broadly categorized into discriminative and generative approaches. Discriminative models such as convolutional neural networks (CNNs) trained with reconstruction losses have been widely adopted (Xu et al., 2020a,b). From the generative modeling perspective, conditional Denoising Diffusion Probabilistic Models (DDPMs) (Ho et al., 2020) have emerged as a recent popular alternative to tackle those observational prediction problems.

Table 1: **Quantitative evaluation of density prediction results reported on the original data scale ($10^{0\sim7}$).** The signs indicate the direction of prediction bias: negative denotes overprediction, and positive indicates underprediction. We **bold** the best results and underline the second-best. $\mu$ and $\sigma$ are the primary criteria for assessment, where values closer to zero indicate superior performance.

| $\rho\,(n_H/cm^{-3})$ | ID | | | OOD | | |
|---|---|---|---|---|---|---|
| Methods | $\mu$ - $p_{\Delta\rho/\rho}$ (↓\|↓\|) | $\sigma$ - $p_{\Delta\rho/\rho}$ (↓\|↓\|) | ME | $\mu$ - $p_{\Delta\rho/\rho}$ (↓\|↓\|) | $\sigma$ - $p_{\Delta\rho/\rho}$ (↓\|↓\|) | ME |
| 2PLF | -2.50±7E-4 | 7.11±2E-3 | -715.95±1.88 | -5.77±3E-4 | 7.21±1E-3 | -1680.82±0.74 |
| 3PLF | -0.77±5E-4 | 4.44±2E-3 | -715.78±1.88 | -3.23±2E-4 | 4.21±7E-4 | -1680.60±0.61 |
| U-NET | -0.25±1E-4 | 1.29±4E-4 | 118.24±0.94 | 5.01±6E-4 | 7.08±1E-3 | 1313.60±0.61 |
| cDDPMs | -0.05±6E-5 | **0.61±6E-4** | 4.61±1.05 | 2.88±5E-4 | 5.33±7E-4 | 1050.51±0.61 |
| Astro-DSB (Ours) | **-0.02±1E-4** | 0.71±3E-4 | 32.09±0.65 | **0.51±2E-4** | 2.32±3E-3 | -5.04±1.32 |

In our experiments, we compare with both conventional and ML-based methods. Specifically for ML, in addition to the state-of-the-art conditional DDPMs, we also re-implement a U-Net (Ronneberger et al., 2015) from CASI-2D (Xu et al., 2020a) as a pixel-to-pixel approach for comparison. Notably, while recent works suggest that generative models offer improved sample quality in image synthesis, their advantage in scientific prediction, especially under physically grounded distribution shift (OOD), remains unclear. Our study explicitly tests this hypothesis by comparing probabilistic and pixel-to-pixel modeling under controlled simulation setups with similar model scales.

**Evaluations.** Our evaluation adheres to established protocols in astrophysical studies, emphasizing **the distributional patterns of relative error between predictions and ground truth values**. Specifically, we assess the normalized relative error (e.g., $\Delta\rho/\rho$), where an ideal prediction should produce a distribution centered around zero. Although this distribution may not strictly follow a Gaussian form, a symmetric shape with a narrow spread around zero indicates accurate and well-calibrated predictions. In contrast, distributions with large bias or heavy tails reflect systematic under- or over-estimation. To quantitatively capture these characteristics, we report the mean $\mu$ (gt - pred) and standard deviation $\sigma$ of the relative error distribution as **primary metrics**. These values provide insights into the central tendency and variability of model performance across different test cases.

In addition to relative error metrics, we also report the Mean Error (ME) between predicted and ground truth values. However, it is important to note that ME interpretation may be affected by the wide dynamic range of physical quantities in astrophysical studies. Specifically, **even a method with a lower ME value does not necessarily indicate superior performance**, as the absolute error can be dominated by regions of large values. For example, a small relative error on a value of $10^7$ can contribute disproportionately to the ME compared to a larger relative error on a value of 10. This imbalance highlights the importance of relative error patterns in accurately assessing performance.

**Implementation details.** To ensure a fair comparison among the tested ML methods, we fix the U-Net backbone architecture across the discriminative U-Net, conditional DDPMs, and our proposed *Astro-DSB*, resulting in approximately 81M learnable parameters for each model. For the U-Net baseline, we train the model for 20 epochs using Mean Square Error (MSE) loss. For conditional DDPMs, we follow the setup from prior works (Xu et al., 2023a, 2025) and adopt a 1000-step schedule with the standard variational lower bound loss (Ho et al., 2020). Our proposed *Astro-DSB* is trained with the same time discretization (1000 steps) using the objective defined in Eq. 5. All models are trained with a batch size of 16 using the AdamW optimizer with a learning rate of 5e-5, running on an AWS cluster with 4 NVIDIA T4 GPUs. Notably, while conditional DDPMs require around 400 epochs to converge, *Astro-DSB* converges in only around 100 epochs under the same setup, reducing overall training time by approximately 75%. This aligns with our expectations and observations in prior vision DSB works, as the target distribution $p_1$ now lies closer to the physically meaningful observation distribution $p_0$, compared to the case where $p_0$ is a zero-mean Gaussian in DDPMs. During inference, the runtime per testing case ranges from approximately 6 to 30 seconds, depending on whether the step skipping (Song et al., 2021) is applied. Notably, we observe no significant difference in prediction performance across these sampler settings in our work.

## 4.3 Experimental results and analysis

**Less biased prediction results for ID and OOD testing.** We present the quantitative evaluation results for density and magnetic field prediction in Tab. 1 and Tab. 2, respectively. Our proposed *Astro-DSB* consistently achieves the best or second-best results across both ID and OOD settings,

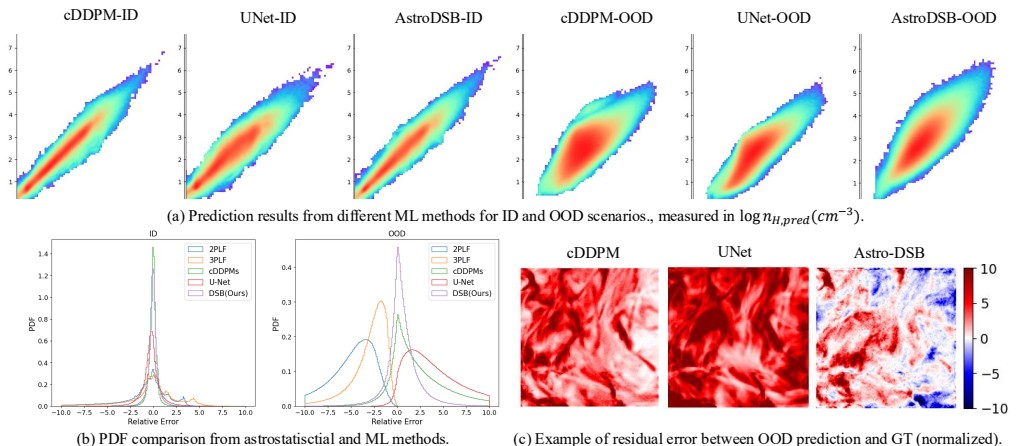

(a) Prediction results from different ML methods for ID and OOD scenarios., measured in $\log n_{H,pred}(cm^{-3})$.

(b) PDF comparison from astrostatisctial and ML methods.     (c) Example of residual error between OOD prediction and GT (normalized).

Figure 3: **Multiple qualitative visualizations for relative error comparisons.** (a) 2D histogram showing the distribution of ground truth (x-axis) versus predictions (y-axis). An accurate and unbiased prediction lies along the 1-to-1 diagonal line. (b) Relative error between the ground truth and predictions. (c) Residual maps showing the difference between model predictions and the ground truth across different methods. Results from *Astro-DSB* present more balanced and unbiased prediction error patterns compared to other ML methods, particularly for OOD testing cases. Best viewed in color and with zoom-in.

Table 2: **Quantitative evaluations of magnetic field prediction results reported on the original data scale ($3E^{1\sim3}$). Bold** values indicate the best results, underlined values indicate the second-best. *Astro-DSB* achieves competitive SOTA with only 25% of the computational cost of cDDPMs.

| $\mathcal{B}\,(\mu G)$ | ID | | | OOD | | |
|---|---|---|---|---|---|---|
| Methods | $\mu$ - $p_{\Delta\mathcal{B}/\mathcal{B}}$ (↓↓↓) | $\sigma$ - $p_{\Delta\mathcal{B}/\mathcal{B}}$ (↓↓↓) | ME | $\mu$ - $p_{\Delta\mathcal{B}/\mathcal{B}}$ (↓↓↓) | $\sigma$ - $p_{\Delta\mathcal{B}/\mathcal{B}}$ (↓↓↓) | ME |
| DCF | 11.80±8E-4 | 21.78±4E-4 | 507.74±0.03 | 5.19±8E-3 | 9.64±3E-3 | 137.76±0.03 |
| PLF | -0.11±5E-5 | 1.23±1E-4 | -1.04±1E-3 | 1.03±9E-5 | 1.47±1E-4 | 17.14±2E-3 |
| U-NET | **-0.06±2E-5** | **0.38±5E-5** | 154.76±0.08 | -0.32±6E-5 | 0.74±7E-5 | 146.38±0.02 |
| cDDPMs | 0.07±2E-5 | **0.38±6E-5** | 123.99±0.04 | **0.12±7E-5** | 0.72±2E-4 | 150.00±0.04 |
| Astro-DSB (Ours) | 0.15±2E-5 | 0.55±5E-5 | 235.85±0.08 | 0.19±6E-5 | **0.67±9E-5** | 131.89±0.02 |

particularly demonstrating robust superiority in OOD predictions, as indicated by the lower mean ($\mu$) and standard deviation ($\sigma$) of the relative error distribution. In contrast, traditional astrostatistical methods such as power-law fitting (PLF) and DCF exhibit large error values, while U-Net and DDPMs suffer from more severe performance degradation under OOD conditions. In particular, the training cost of *Astro-DSB* is only 25% of the previous SOTA conditional DDPMs method.

Fig. 3 further highlights the superior generalization of *Astro-DSB*, with sharply peaked PDF curves and coherent residual error patterns, especially in OOD scenarios. In comparison, both machine learning methods, U-Net modeling and cDDPMs, tend to underpredict in OOD testing, while conventional methods, including 2PL and 3PL, usually overpredict for density estimations in unseen simulations. These results demonstrate that *Astro-DSB* offers robust and efficient predictions for astrophysical inverse tasks. Extended evaluations can be found in the appendices.

**Extended robustness on observations from Taurus B213.** We further test our *Astro-DSB* on the Herschel column density map of Taurus B213 (Li and Goldsmith, 2012; Palmeirim et al., 2013) using the model trained purely on synthetic simulations. Fig. 4 visualizes the observed column density map alongside our predicted volume density map. Despite the lack of pixel-level ground truth in real-world observations, our predictions demonstrate a coherent spatial structure and effectively capture the filamentary features of B213, as qualitatively verified by astrophysicists. Notably, the results align well with single-pointing constraints derived from indirect measurements using the density tracer $HC_3N$ (Li and Goldsmith, 2012), which suggests that *Astro-DSB* can generalize effectively beyond simulated scenarios, providing reliable physical predictions even on real observational data. More experimental results are in the appendices.

Table 3: **Ablation results for the key methodology designs on density predictions.**

| $\rho\,(n_H)$ | ID | | | OOD | | |
|---|---|---|---|---|---|---|
| Method | $\mu$ - $p_{\Delta\rho/\rho}$ ($\downarrow\downarrow$) | $\sigma$ - $p_{\Delta\rho/\rho}$ ($\downarrow\downarrow$) | ME | $\mu$ - $p_{\Delta\rho/\rho}$ ($\downarrow\downarrow$) | $\sigma$ - $p_{\Delta\rho/\rho}$ ($\downarrow\downarrow$) | ME |
| w/o M1 & M2 | -0.19±3E-4 | 1.75±0.04 | 66.54±1.49 | 4.24±2E-3 | 14.71±0.12 | 1086.66±0.66 |
| w/o M1 | **-3E-3±7E-5** | **0.65±6E-4** | 35.86±0.44 | -1.84±5E-4 | 5.06±4E-3 | -2297.48±1.10 |
| w/o M2 | 7E-3±1E-4 | 1.63±5E-4 | 37.15±1.53 | -14.66±5E-3 | 54.95±3.47 | -12069.44±9.44 |
| Astro-DSB (Full) | -0.02±1E-4 | 0.71±3E-4 | 32.09±0.65 | **0.51±2E-4** | **2.32±3E-3** | -5.04±1.32 |

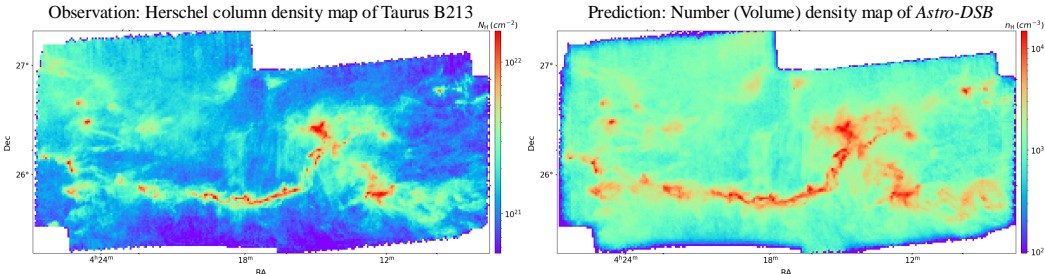

Figure 4: **Visualization of observables and our predicted results on Herschel column density map of Taurus B213**, shown in the plane of the sky with the x-axis representing right ascension and the y-axis representing declination. The observable map is measured in $N_H/cm^{-2}$, and the prediction is in $n_H/cm^{-3}$. While there is no ground truth for real-world observations, our prediction results are consistent with those obtained with other independent approaches.

**Ablation studies on key methodological components.** We evaluate the effectiveness of two key components in our *Astro-DSB* design: **M1: Noise alignment** and **M2: Observable enhancement**, by systematically removing them and reporting the results in Tab. 3. Our findings reveal that the absence of either module significantly degrades performance, with **M2** being particularly effective for cases of OOD and stable prediction indicated as indicated by $\sigma$ values. The full *Astro-DSB* achieves the best balance of prediction accuracy and generalization robustness.

## 5 Conclusion and Discussion

In this work, we address a class of astrostatistical inverse prediction tasks by introducing *Astro-DSB*, a dynamic generative modeling framework based on the Diffusion Schrödinger Bridge. Targeting key physical quantities within GMCs, *Astro-DSB* learns the probabilistic transition between partial observables and true physical states, offering a more flexible alternative to conventional one-step or purely discriminative approaches. Our study reveals that, in contrast to static tasks where dynamic modeling may appear excessive, explicitly learning distributional transitions is both meaningful and beneficial in physically grounded systems demonstrated via improved generalization in out-of-domain simulations and real-world observables. Overall, *Astro-DSB* offers better alignment with physical interpretability and supports the broader potential of probabilistic modeling in scientific discovery.

**Limitation and future direction.** Despite the improved robustness and performance to unseen simulation conditions, our method does not explicitly model the full forward physical equations that govern the GMC evolution. While this is reasonable given the intractable nature of observational inversion problems in this work, future extensions could involve integrating multiple intermediate states, leveraging richer supervision signals, or directly encoding the dynamic physical laws into the generative modeling process to further strengthen the alignment between ML and physical dynamics for extended applications with well-defined physical forward processes.

## Acknowledgments and Disclosure of Funding

This research was partially supported by an Amazon Research Award to OR and YZ. Any opinions, findings, and conclusions or recommendations expressed in this material are those of the author(s) and do not necessarily reflect the views of Amazon. DX acknowledges the support of the Natural Sciences and Engineering Research Council of Canada (NSERC), [funding reference number 568580]. DX also acknowledges support from the Eric and Wendy Schmidt AI in Science Postdoctoral Fellowship Program, a program of Schmidt Sciences. YZ also acknowledges the travel funding support by the French National Research Agency (ANR) via the "GraspGNNs" JCJC grant (ANR-24-CE23-3888), coordinated by Johannes F. Lutzeyer from École Polytechnique.

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

## Technical Appendices and Supplementary Material

A statement of broader impacts is included in Section A. Section B provides additional details on the relevant astrophysical background for readers interested in research around Giant Molecular Clouds (GMCs) and star formation. Further experimental details, including extended qualitative comparisons across different methods, are presented in Section C. We have also included our **core code** as part of the supplementary material, with the full codebase and datasets to be released upon acceptance.

## A    Statement of broader impacts

We apply the Diffusion Schrödinger Bridge framework to observational inverse prediction problems in astronomy. We do not anticipate any immediate societal risks or ethical concerns associated with this research. Our work focuses on advancing scientific understanding in a domain with minimal direct human impact, and all data used in this study are derived from publicly available astronomical observations or licensed simulations.

Beyond its immediate scientific goals, this research contributes positively to the broader advancement of interdisciplinary science. By demonstrating the application of a principled generative modeling framework in a physically grounded domain, our work helps bridge the gap between data-driven machine learning and theory-driven physical modeling. This approach not only promotes interpretability and robustness in ML research but also supports the long-term vision of using AI to accelerate scientific discovery. We hope this study encourages further collaboration between the machine learning and astrophysics communities, especially in leveraging modern ML techniques to understand complex natural systems.

## B    Astrophysical background

We provide extended details about GMCs and star formation research for readers interested in the astrophysical background related to this work. In this work, we focus on studying the physical conditions and processes within giant molecular clouds (GMCs), by inferring the precise distributions of various critical physical quantities from limited observational data.

Giant Molecular Clouds (GMCs) are key structures within the interstellar medium (ISM) of galaxies, composed primarily of cold gas and dust. The ISM, which fills the space between stars, also contains cosmic rays and magnetic fields and plays a central role in galactic evolution by regulating star formation rates (Spitzer Jr, 2008). GMCs are of particular interest because they harbor most of the dense gas in the ISM, which is essential for the formation of stars (McKee and Ostriker, 2007). The physical conditions within GMCs are highly variable and complex, exhibiting spatial and temporal variations in density, temperature, velocity, and magnetic field strength and orientation. These factors give rise to a wide array of astrophysical phenomena, including the formation of protostars (Heyer and Dame, 2015), star clusters (Krumholz et al., 2019), and complex organic molecules (Bisbas et al., 2023). Among all physical quantities, we study two fundamental quantities in this work, which are the gas density ($\rho$) and the magnetic field ($\mathcal{B}$).

### B.1    Gas density

The gas density (i.e., mass per unit volume), often measured as the number density of hydrogen nuclei (H), is considered one of the most fundamental quantities in GMC studies. Under the assumption of an abundance of one He nucleus for every 10 H nuclei in interstellar gas, we have a mass per H nucleus of $\mu_H = 1.4 m_H = 2.34 \times 10^{-24} g$, leading to $n_H = 1 cm^{-3}$ equivalent to $\rho = 2.34 \times 10^{-24} g\, cm^{-3}$.

The gas density directly governs gravitational collapse, sets the local timescale for star formation, and strongly influences the thermal and chemical evolution of the gas. For example, denser regions within GMCs are more likely to become gravitationally unstable, leading to the formation of stars or star clusters. Moreover, the density controls the rates of radiative cooling and chemical reactions, such as molecule formation and dust grain processes, which are crucial for understanding the lifecycle of interstellar gas and the conditions for planet formation. Accurately inferring the spatial distribution of density within GMCs is therefore essential for modeling the early stages of stellar evolution.

In astronomical studies, it is usually very challenging to precisely quantify the number density of GMCs from observations. The traditional astrostatistical approach relies on the observations of column density and certain assumptions on the geometry of the clouds. For example, a cylindrical geometry for filamentary structures or spherical geometry for dense cores (Palmeirim et al., 2013). Bisbas et al. (2021, 2023) later propose an empirical power law to convert the observed column density to the mean number density of GMCs based on MHD simulations (Wu et al., 2015), which has also been adapted in our work as two baseline methods (i.e., two-component power-law conversion and three-component power-law conversion). Despite being effective in similar simulation data, the prediction performance of these methods often suffers from a significant drop when new observations present a large distributional shift compared to the original fitting data (i.e., OOD testing cases). In our work, we aim to introduce a machine learning based approach with better robustness and generalization ability.

## B.2 Magnetic field

Magnetic fields, often measured in microgauss ($\mu G$), represent another critical physical quantity within the interstellar medium (ISM), playing a fundamental role in shaping the structure and regulating the dynamics of GMCs (Crutcher, 1999, 2012). They can provide support against gravitational collapse, guide anisotropic gas flows, and influence the morphology of star-forming regions through magnetic tension and pressure. In combination with turbulence, magnetic fields are believed to significantly affect the efficiency and fragmentation of star formation across spatial scales.

Similar to the previous density estimation, estimating the magnetic field strength within GMCs is a challenging task. The Davis–Chandrasekhar–Fermi (DCF) method is a widely adopted technique in astrophysics to estimate the plane-of-sky magnetic field strength $\mathcal{B}_{POS}$ in molecular clouds based on observed dust polarization (Davis, 1951; Chandrasekhar and Fermi, 1953; Beck, 2016), which relates the magnetic field strength to the polarization angle and other observable properties. Its core assumption is that an equipartition exists between the magnetic field energy and the turbulent kinetic energy of the gas. Specifically, the DCF method states that the relation between the gas density $\rho$, the nonthermal velocity dispersion $\sigma_V$, and the polarization angle dispersion $\sigma_{PA}$ gives the POS magnetic field strength $B_{POS}$ as:

$$B_{POS} = f\sqrt{4\pi\rho}\frac{\sigma_V}{\sigma_{PA}}, \tag{7}$$

where $f$ is a correction factor accounting for projection and geometric effects. More recently, a variant of DCF method (Skalidis and Tassis, 2021) takes the compressible modes into account and modifies the previous Eq. 7 to:

$$B_{POS} = \sqrt{2\pi\rho}\frac{\sigma_V}{\sqrt{\sigma_{PA}}}. \tag{8}$$

While widely used, this method suffers from several limitations: (1) the equipartition assumption may not hold in strongly magnetized or shock-dominated regions; (2) $\sigma_\phi$ becomes unreliable in regions with low signal-to-noise ratio or complex field morphology; and (3) accurate density and turbulence estimates are required for robust magnetic field inference, which are themselves uncertain. Moreover, DCF provides only an average value over large regions and lacks the ability to resolve fine-scale spatial structures.

In this work, we aim to bypass the limitations of traditional analytic approximations by leveraging machine learning to directly infer the underlying magnetic field distribution from observables, jointly with other key physical quantities such as gas density. Our proposed framework thus offers a more flexible and data-driven solution, especially under conditions with limited observability or strong distribution shifts.

In Fig. 5, we visualize an example of molecular cloud data used in machine learning methods from Xu et al. (2025), including the map of column density with magnetic field directions, the magnetic field angle dispersion, the LOS velocity dispersion, and the projected magnetic field strength.

## C  Additional experimental results and analysis

We provide extended details about the experiments in this section.

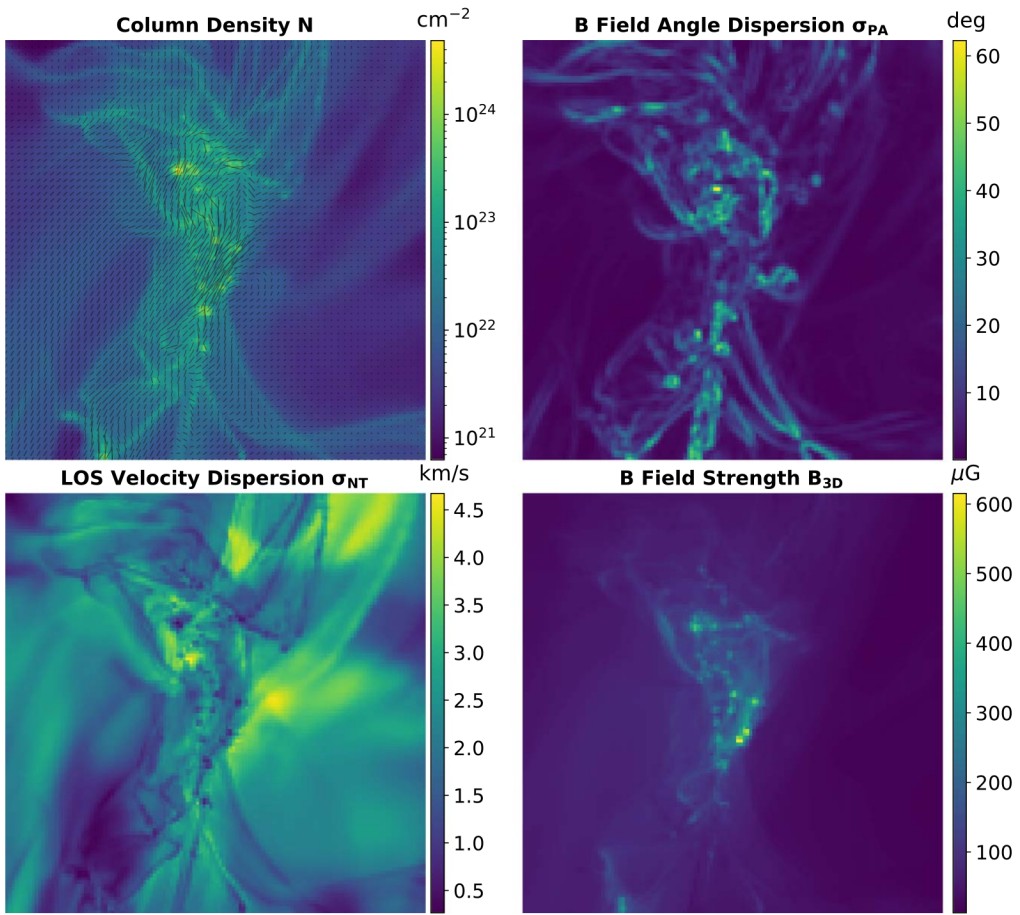

Figure 5: **Illustration of different synthetic observables from (Xu et al., 2025) for density and magnetic field estimation.** Top left: map of column density with magnetic field directions; Top right: magnetic field angle dispersion; Bottom left: LOS velocity dispersion; Bottom right: projected magnetic field strength.

### C.1 More details about the magnetohydrodynamics data simulations

We follow the same simulation setups used in prior works that apply conditional DDPMs to astrophysical observational prediction (Xu et al., 2023b, 2025; Wu et al., 2020; Hsu et al., 2023). While the design and analysis of physical simulations represent an important research direction on their own that is distinct from the machine learning focus of this work, we provide only the essential background here and refer interested readers to the original astrophysical literature for further details.

**Density simulation.** For the density simulations, we adopt the magnetohydrodynamic (MHD) simulations of colliding and non-colliding GMCs following prior works (Wu et al., 2020; Hsu et al., 2023), performed using the adaptive mesh refinement (AMR) code Enzo (Bryan et al., 2014). The simulations include self-gravity, magnetic fields, and thermal processes under a photodissociation radiation field ($G_0 = 4$ Habings) and cosmic-ray ionization rate $\zeta = 10^{-16} \, \mathrm{s}^{-1}$. Two clouds with radii of 20 pc are initialized in a $128^3 \, \mathrm{pc}^3$ domain with a $256^3$ grid resolution.

Each cloud starts with $n_H = 83 \, \mathrm{cm}^{-3}$ and $T = 15$ K, while the ambient gas is set at $n_H = 8.3 \, \mathrm{cm}^{-3}$ and $T = 150$ K to maintain pressure balance. A multiphase temperature structure develops over time, with high-density regions ($n_H \gtrsim 10^3 \, \mathrm{cm}^{-3}$) cooling to 10–20 K, and low-density regions ($n_H \lesssim 10 \, \mathrm{cm}^{-3}$) reaching up to 1000 K. Magnetic fields are initialized at 10–50 $\mu$G, oriented $60°$ to the collision axis. The refinement strategy ensures that the local Jeans length is resolved by at least eight cells.

The simulation runs span 5 Myr, with snapshots taken at 3 and 4 Myr. Column density maps and corresponding line-of-sight (LOS) mass-weighted number density maps are extracted at multiple physical scales (32, 16, 8, and 4 pc), and projected to $128 \times 128$ pixel images. The resulting dataset contains 7179 samples, with an approximate 80/20 split between training and testing. For more details, please refer to (Wu et al., 2020; Hsu et al., 2023).

**Magnetic field simulation.**   For magnetic field prediction tasks, we adopt ideal magnetohydrodynamics (MHD) simulations following the setup in prior work (Wu et al., 2020; Hsu et al., 2023), using the MUSCL-Dedner method and HLLD Riemann solver within the AMR code ENZO (Bryan et al., 2014). These simulations incorporate self-gravity, magnetic fields, and thermal processes under a photodissociation radiation field with $G_0 = 4$ and a cosmic-ray ionization rate of $\zeta = 10^{-16} \, \text{s}^{-1}$. Each simulation initializes two clouds (radius 20 pc) in a $128 \, \text{pc}^3$ domain at $256^3$ resolution, with initial hydrogen number density $n_\text{H} = 83 \, \text{cm}^{-3}$, temperature $T = 15 \, \text{K}$, and solenoidal turbulence satisfying $v_k^2 \propto k^{-4}$ for $2 \leq k \leq 20$. The magnetic field is initialized at a $60°$ angle to the collision axis with strengths of 10, 30, and $50 \, \mu\text{G}$ across different cases, and four additional refinement levels are used to resolve the local Jeans length. For each magnetic field configuration, we model both colliding and non-colliding GMC scenarios, where colliding clouds are offset by $0.5R_\text{GMC}$ and have a relative velocity of $10 \, \text{km/s}$. These simulations run up to $4.1 \, \text{Myr}$ and capture early evolutionary phases before star formation begins.

To generate realistic observables, we compute maps of column density, line-of-sight (LOS) mass-weighted polarization angle, LOS nonthermal velocity dispersion, and projected 3D magnetic field strength across varying AMR levels. The final dataset consists of 25,479 images at $128 \times 128$ resolution, sampled across physical scales of 32, 16, 8, and 4 pc. To enhance diversity, we include multiple projection angles and aggregate LOS effects without manually disentangling contributions such as turbulence or cloud-scale motions, consistent with observational limitations.

## C.2   Additional training details

As we have mentioned in the main paper, our proposed *Astro-DSB* benefits from a faster convergence speed and thus less computational resources compared to conditional DDPMs and vanilla DSB models. In Fig. 6, we compare the training curves among different variants of DSB as additional ablation analysis.

Compared to previous works that leverage conditional DDPMs as the prediction framework (Xu et al., 2023a, 2025), our *Astro-DSB* converges around 25k iterations as shown in Fig. 6(d), which corresponds to approximately 10 GPU hours. In contrast, conditional DDPMs take around 40 GPU hours to train.

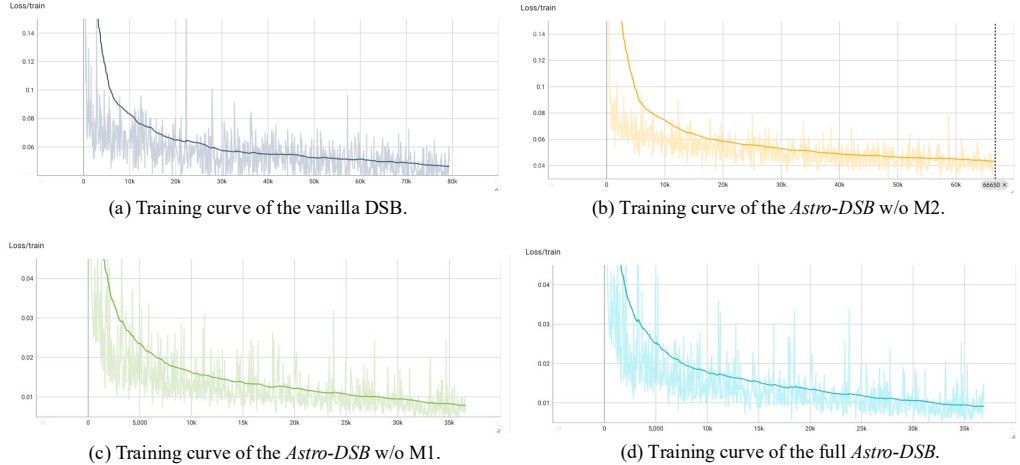

(a) Training curve of the vanilla DSB.

(b) Training curve of the *Astro-DSB* w/o M2.

(c) Training curve of the *Astro-DSB* w/o M1.

(d) Training curve of the full *Astro-DSB*.

Figure 6: **Training curve comparison of different variants of DSB for ablation analysis.** We show that the proposed **M2** component on the observable enhancement contributes to a faster training convergence speed. Best viewed in color and with zoom-in.

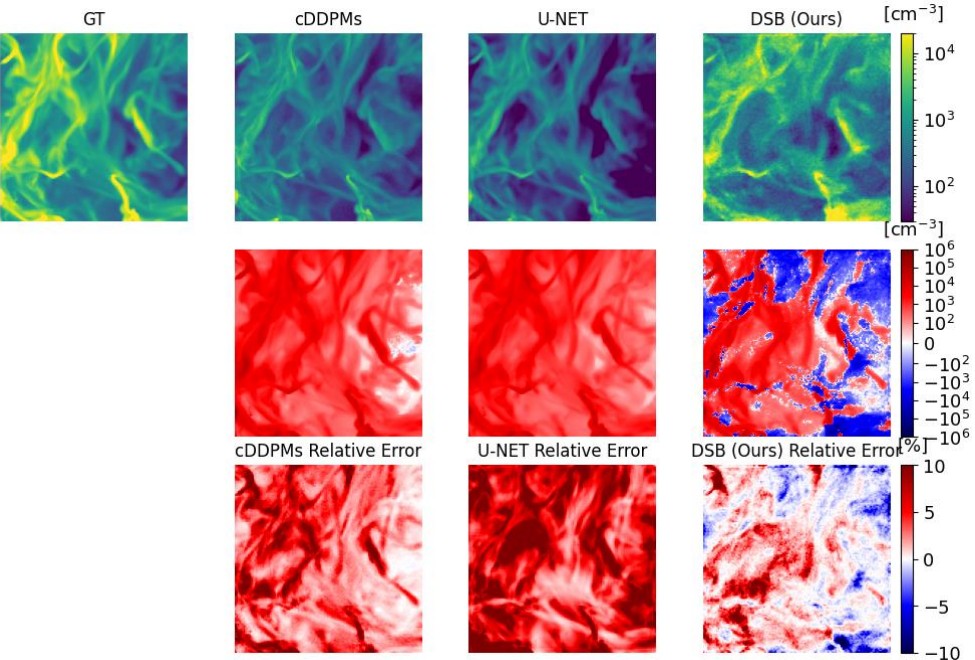

Figure 7: **Additional visualization of relative error patterns from different machine learning methods for the density prediction task on OOD testing cases.** In the first row, we qualitatively show the GT and prediction results from conditional DDPMs, U-Net, and our proposed *Asro-DSB* method. The second and third rows present the relative errors computed by "GT - pred" in absolute values and the values after log transformation, respectively. Compared to other ML methods that tend to underpredict in OOD testing cases, our *Astro-DSB* achieves less biased and more robust prediction performance.

### C.3 More details about experiments on simulation data

As explained in the main paper, evaluations in astrophysical prediction particularly emphasize the relative error patterns exhibited between the prediction results and the ground-truth distributions. In Fig. 7, we present additional visualization of relative error patterns from different machine learning methods for the density prediction task on OOD testing cases. The first row shows the ground truth and prediction results from conditional DDPMs, U-Net, and our proposed *Astro-DSB*. The second and third rows visualize the corresponding relative errors, computed as "GT - Pred" in absolute values and after applying log transformation, respectively. Compared to DDPMs and U-Net, which tend to significantly underpredict dense structures in the OOD regime, *Astro-DSB* produces more balanced and spatially consistent residuals. These visual results further support the claim that our method achieves improved robustness and lower bias when generalizing beyond training distributions.

To better understand the impact of each design component, we compare the distribution of relative errors for different methods in Fig. 8. Traditional power-law baselines (p2, p3) and vanilla DDPMs (cdm) exhibit large biases and heavy tails, especially under distribution shifts. The vanilla DSB (psbv1) slightly reduces error spread, while our proposed components—M1 and M2—each contribute to sharpening and centering the PDF. The full model (psbv4) demonstrates the narrowest and most symmetric error distribution, highlighting its robustness and well-calibrated behavior in OOD settings.

### C.4 More details about experiments on real observations

We further compare our proposed *Astro-DSB* with previous machine learning methods on real observational data from the Taurus B213 region, focusing on the inferred number density distribution. As shown in Fig. 4, the prediction of *Astro-DSB* closely matches the spatial structures and filament morphology present in the original Herschel column density map. In contrast, the prediction from

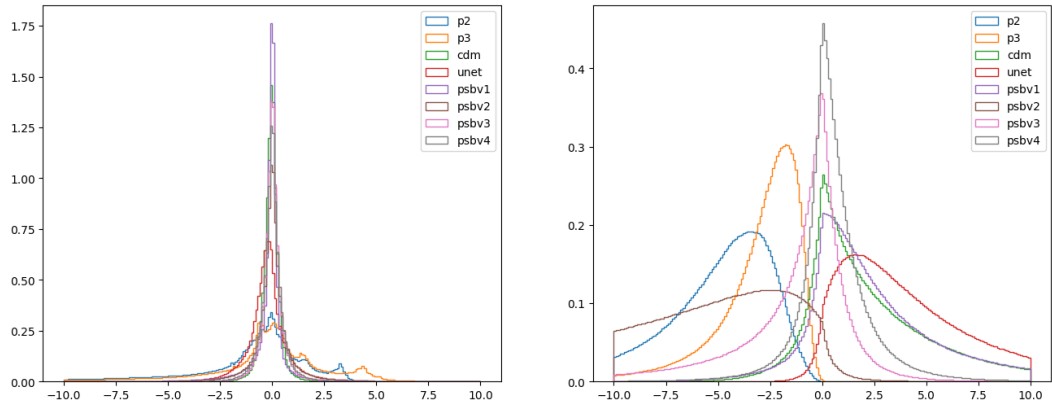

Figure 8: **PDF comparisons across different methods for the density prediction task on ID and OOD data.** Methods *p2* and *p3* correspond to two- and three-component power-law conversions, respectively. *cdm* denotes conditional DDPMs. *psbv1* is the vanilla diffusion Schrödinger bridge without our proposed modules. *psbv2* and *psbv3* remove M1 (noise alignment) and M2 (observable enhancement), respectively, while *psbv4* represents the full *Astro-DSB* model.

CASI-2D (Xu et al., 2020a) fails to capture fine-scale density variation and underestimates extended dense regions. The conditional DDPMs produce more coherent predictions but still smooth out localized dense structures. Overall, *Astro-DSB* exhibits better spatial continuity, structure recovery, and dynamic range, demonstrating its superior robustness and generalization capability on real-world astrophysical observations.

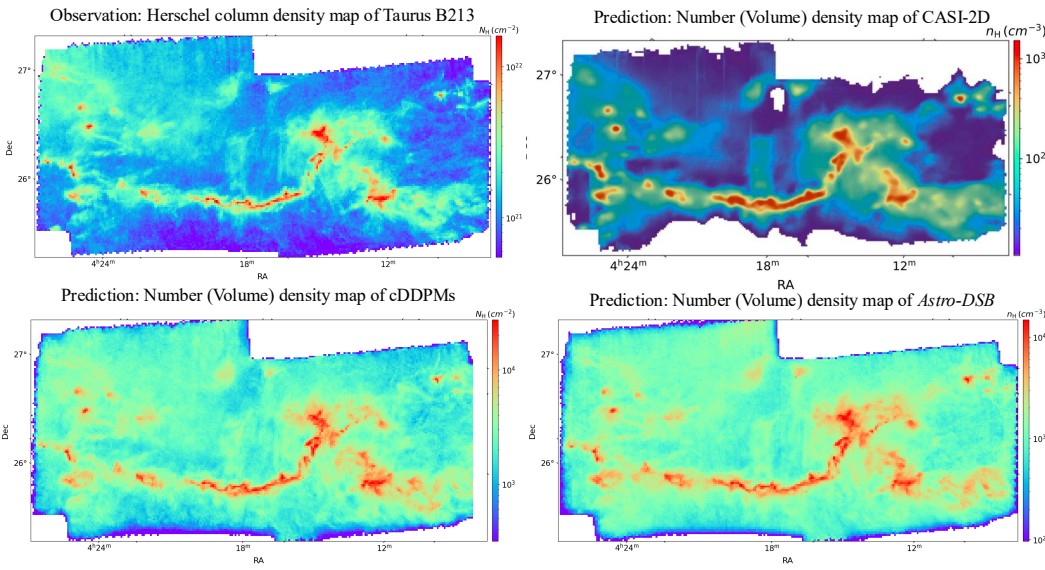

Figure 9: **Additional qualitative comparisons among different machine learning methods on Taurus B213 number density prediction.** CASI-2D (Xu et al., 2020a) is a previous ML method with U-Net architecture trained with discriminative reconstruction loss.

