# OpenReview forum: "Dynamic Diffusion Schrödinger Bridge in Astrophysical Observational Inversions"
_NeurIPS.cc/2025/Conference — NeurIPS 2025 poster_

### Official Review · Reviewer_fbj1 · 2025-06-01

**Clarity:** 2
**Significance:** 3
**Originality:** 2
**Rating:** 4
**Confidence:** 3

**Summary:**

In this paper, the authors propose to use a Bridge Matching procedure in order to astrostatistical inverse problems within iant Molecular Clouds. Their method is similar to I2SB [1]. The authors introduce a few key changes however which improve the methods, especially on OOD tasks. More precisely, they assume a modelling noise which is Gaussian and also further condition the model based on the clean observation $y$ similarly to [2]. Finally, as the data in astrostatistical setting is extremely large, they proceed using patch based training and an aggregation method. They conduct their evaluation on two key physical processes from magnetohydrodynamics simulations. They show improvements over conditional diffusion model baselines especially in OOD settings.

[1] I^2SB: Image-to-Image Schrödinger Bridge (2023) -- Liu et al.

[2] Augmented Bridge Matching (2023) -- De Bortoli et al.

**Questions:**

L.8 “our proposed paired DSB method” > We would like to take the opportunity to clarify that “paired DSB” is non-sensical as DSB and any Schrodinger Bridge and Optimal Transport (OT) based method must work on unpaired data as one of the goal of OT is to discover said pairing. We are aware of some confusion in the literature but we feel like this misunderstanding should be properly addressed.

L.28 “we argue that the true power of this reformulation lies in the fact that it allows for diffusion models to capture the underlying data dynamics” > There are other benefits of OT compared to diffusion models especially since they work in unpaired settings as highlighted above.

l.167  > I2SB is not a special case of the Schrodinger Bridge problem.

L.174 > In M1, compared to I2SB the authors introduce another noise variable in $y_1 - \varepsilon$. Does this $\varepsilon$ correspond to the one introduced in (1), i.e., when specifying the perturbation model. It seems to be the case reading l.178. How realistic is this assumption?

L.183 > The proposed method in M2 has already been proposed in the literature in [1]. The only difference between [1] and the proposed model is that [1] would take $y - \varepsilon$ as an input while the current model takes $y$ as an input.

L.189  “framework and helps simultaneously obtain lower loss values with a faster convergence speed” > It is unclear to me if the claim of lower loss value is grounded or not as a model that takes $y$ as a conditioning will have more parameters than a model that doesn’t. I guess, the main question is: “Is the improvement in performance observed by the authors only due to the increased number of parameters or is it due to the modeling choices?” My guess is that the modeling choices matter however, I do not think that the current experiments are enough to support such a claim.

L.202 > I think it would be very interesting to describe in more details the aggregation strategy of AstroDSB.

L.214 > I am by no means an expert in astrophysics and therefore will provide comments based on a machine learning point of view here. However, it seems that 5707 training samples in the molecular density casse and 19100 training samples in the magnetic field strength case is way less than the size of classical datasets used in machine learning. I have a few questions there: 1) is the size of the training dataset considering the patch-based augmentation? 2) since the dataset is quite small, are there safeguards in place to show that the model does not memorize the example ground-truths?

L.204 > While describing the astrophysical data, even though the authors give the size of the patch it would be interesting to also give the size of the molecular density and the magnetic fied strength, i.e., how may patches do I need to reconstruct the data?

L.267 > It seems that the main method of evaluation in this work is to report the mean and standard deviations of the relative errors. The authors write an interesting paragraph about the limitation of the Mean Error (ME). Is it a novel observation or is it well-known in the literature?

L.274 > While investigating the implementation details it seems that the method used by the authors rely on quite outdated methods. Notably using a time discretization of 1000 steps. I would like to encourage the authors to investigate recent works such as [2,3] to improve their results.

Figure 3 (b) is extremely hard to parse.

[1] Augmented Bridge Matching (2023) – De Bortoli et al.

[2] Simpler Diffusion (SiD2): 1.5 FID on ImageNet512 with pixel-space diffusion (2024) – Hoogeboom et al.

[3] Simple ReFlow: Improved Techniques for Fast Flow Models (2024) – Kim et al.

**Ethical Concerns:**

["NO or VERY MINOR ethics concerns only"]

**Limitations:**

The authors have explicitly addressed the limitations of their work in the conclusion.

**Quality:**

3

**Strengths And Weaknesses:**

Strength:

* The use of I2SB techniques for astrostatistical problem is new.

* The results regarding the differences with conditional diffusion models are quite compelling.

Weaknesses:

* There seems to be some confusion around methodological and technical terms (see below)

* (Minor) It is not clear to me what is the significance of those results in the astrostatistical community as I am not an expert on the topic.

---

> ### Author Rebuttal · Authors · 2025-07-30
>
> We greatly appreciate the reviewer’s careful and thorough suggestions regarding the terminology clarifications. We will incorporate these writing suggestions to ensure greater clarity and rigor. Below, we provide detailed responses to the specific questions raised.
>
> ---
> **L.174 -> In M1, compared to I2SB the authors introduce another noise variable in $y - \lambda$. Does this correspond to the one introduced in (1), i.e., when specifying the perturbation model. How realistic is this assumption?**
>
> **A1:** Yes, the noise variable $\lambda$ corresponds to $\varepsilon$ introduced in Eq. (1), and this is a **realistic and widely adopted assumption**. In **existing astro literature**, noise is commonly introduced both by the instrumentation (the telescope) and by foreground/background gas emission with a signal-to-noise ratio (SNR) at the order of 10.  For instance, the noise level from the column density map from the Five College Radio Astronomical Observatory (FCRAO) based on the 12CO observations [1] is around 0.28K, v.s. signal of 1-7K. Similarly, the extinction maps derived from Spitzer infrared observations toward several molecular clouds [2,3] shows ~10% uncertainties.
>
> ---
> **L.189 -> Is the improvement in performance observed by the authors only due to the increased number of parameters or is it due to the modeling choices?**
>
> **A2:** We appreciate the thoughtful question. To further examine whether the observed performance gain is solely due to the modeling choice, we conducted an additional ablation where the conditioning inputs were replaced with zero-padding to ensure the number of parameters remained the same. The results remain consistent with those reported in the original manuscript, supporting our claim that the improvement **stems from the modeling formulation rather than increased capacity**. We are happy to include these new results in the revised version, and provide the corresponding curve as those in Figure 6 from Appendix C.2 as rigorous validation (as we are not able to provide additional figures in the rebuttal this year).
>
> | cond | ID-$\mu$ ($\downarrow$) | ID-$\sigma$ ($\downarrow$) |  ID-ME | OOD-$\mu$ ($\downarrow$) | OOD-$\sigma$ ($\downarrow$) | OOD-ME |
> |:--------:|:--------:|:-----------:|:------:|:---------:|:------------:|:------:|
> |   ours   |   -0.02  |     0.73    |  32.09 |    0.51   |     2.32     |  -5.04 |
> |   void   |   -0.10   |     2.12    | 62.72 |    6.16   |     12.23     | 381.70 |
>
> ---
> **L.202 -> Would be very interesting to describe in more detail the aggregation strategy of AstroDSB.**
>
> **A3:** The aggregation strategy in Astro-DSB is designed for compatibility with high-resolution real-world observations. Specifically, given a large observational map $y_{obs}$, we first perform a normalization step as preprocessing to convert it into a normalized input space. We then partition the normalized map into overlapping patches $y_{obs}^i$ using a fixed step size $k$, which is conceptually similar to the stride in convolutional neural networks. For example, if the patch size is $128×128$ and step $k = 8$, the top-left coordinates of successive patches shift by 8 pixels.
>
> Each patch is then independently passed through the trained model to obtain corresponding predictions $\tilde{x}_1^i$. Finally, the overlapping predicted patches are aggregated by averaging over overlapping regions, producing a full-resolution output map aligned with the input observation. This strategy enables scalable and accurate inference on large sky surveys without introducing padding artifacts. We are happy to further clarify this process in the revised manuscript if helpful.
>
> ---
> **L.214 -> 1) Is the size of the training dataset considering the patch-based augmentation? 2) Since the dataset is quite small, are there safeguards in place to show that the model does not memorize the example ground-truths?**
>
> **A4:** Thank you for the questions.
>
> - The reported training dataset size does not include the patch-based augmentation.
>
> - Regarding memorization: this is less of a concern in our setting compared to conventional ML, and does not appear to be an issue based on our OOD results.
>
>    - *Why is memorization less relevant in our inverse prediction task?* Unlike image generation tasks where memorization can lead to privacy or copyright concerns, the goal here is to recover physical quantities grounded in simulation-based truth. The focus is on whether the model can generalize the underlying mapping from observation to physical state driven by implicit physical dynamics, rather than replicate specific inputs.
>
>    - *Why are we confident the model isn’t memorizing?* Our OOD test cases are drawn from very different simulations (with distinct dominant physical processes) or real observations (Taurus B213), showing strong distributional shifts from training data. If the model were merely memorizing training examples, both discriminative ML baselines and classical astrophysical methods should perform similarly well, but the results show otherwise.
>
> ---
> **L.204 -> It would be interesting to also give the size of the molecular density and the magnetic field strength, i.e., how many patches do I need to reconstruct the data?**
>
> **A5:** This question touches on key aspects of both the training data and observational setup in astronomy. As a briefsummary, the number of patches needed depends primarily on the size of the observational map and the target resolution.
>
> - **In real observations**, typical molecular clouds range from 10 to 300 light-years. For instance, the Taurus GMC spans roughly 78 light-years. The test case used in our study focuses on a specific region within the Taurus B213 filament, which is approximately 32 light-years in length.
>
> - **In practical studies**, including our ML setup, we used a physical resolution of 0.12 light-years per pixel at the distance of Taurus, which aligns well with the resolution range in our training data. It’s important to note that telescopes provide angular resolution, not physical resolution. To convert to physical resolution, we multiply the angular resolution by the distance to the cloud. As a result, more distant clouds will have coarser physical resolution, even with the same angular resolution.
>
> - **As for visualization purposes**, our Taurus B213 map shown in the paper has a pixel resolution of 128×264. Following our aggregated sampling as detailed in **A3**, we used 18 patches to reconstruct it.
>
> ---
> **L.267 -> The authors write an interesting paragraph about the limitation of the Mean Error (ME). Is it a novel observation or is it well-known in the literature**
>
> **A6:** Yes, this is a rather novel observation in the ML context, yet critical in scientific domains like astrophysics where data often spans an extremely wide dynamic range (e.g., 10^{0} to 10^{7} in the case of density). This limitation of ME is less commonly discussed in standard ML and computer vision tasks, where inputs like RGB images are naturally bounded in [0, 255], and such scale imbalances are rare. In contrast, astrophysical data, particularly when unnormalized and interpreted in its physical units, exhibits significant value imbalances that can distort ME interpretation. We believe highlighting this issue is particularly relevant for applying ML to real-world scientific data.
>
> ---
> **L.274 -> Encourage the authors to use more recent works, such as Simpler Diffusion (SiD2) and Simple ReFlow to improve results.**
>
> **A7:** We appreciate the suggestion and are aware of the fact that many recent ML works aim to improve image synthesis via updated training recipes, advanced schedulers, or sampler designs, including those mentioned. In our experiments, we did try alternative inference techniques with skipped diffusion steps, as noted in L.289: “We observe no significant difference in prediction performance across these sampler settings in our work.” Regarding training recipes, it seems that there is no publicly available code for SiD2 or Simple ReFlow, and thus, we were unable to reproduce them during the rebuttal phase. That said, we still thank the reviewer for the recommendation and consider these valuable directions for future exploration.
>
> ---
> **On the significance of this ML work in the astrostatistical setting.**
>
> **A8:** The significance of our work can be understood from both the scientific importance of the tasks and the methodological value.
>
> - **Scientific relevance**: Understanding the physical conditions of GMCs is central to the study of star formation, which is an area of active and long-standing interest in the astrophysics community (e.g., see references McKee and Ostriker, 2007; Larson, 2003; Madau and Dickinson, 2014; Xu, 2023, 2025 and others in our original manuscript). Accurate inference of physical quantities from observational data is essential to uncovering the initial conditions that govern how stars form and evolve.
>
> - **Methodological contribution**: Traditional astrostatistical methods rely on simplified assumptions and linear approximations, while existing machine learning approaches often focus on pointwise regression or generative priors that lack physical grounding. Our proposed DSB framework improves upon both by modeling distribution-level transitions between physical states and observations. This enables better generalization across different simulation setups and robustness when applied to real observational data, as demonstrated in our OOD and Taurus B213 results.
>
> ---
> **For other suggestions:** We sincerely appreciate the detailed feedback and would be happy to incorporate these suggestions into the revised manuscript.
>
> ---
> [1] Large-Scale Structure of the Molecular Gas in Taurus Revealed by High Linear Dynamic Range Spectral Line Mapping, APJ 2008
>
> [2] Spectroscopic Infrared Extinction Mapping as a Probe of Grain Growth in IRDCs, APJ 2015
>
> [3] Denoising Diffusion Probabilistic Models to Predict the Density of Molecular Clouds, APJ 2023

---

> ### Author Response · Authors · 2025-08-04
>
> We sincerely appreciate the reviewer’s response and value the opportunity to engage in further discussion on the remaining concerns.
>
> > I would suggest to increase the size of the critical path network in the unconditional case so that the number of active parameters match the ones of the conditional ones.
>
> **A1:** Thank you for the thoughtful suggestion. In response, we conducted an additional set of controlled experiments by slightly expanding the core UNet architecture in the unconditional case to match the parameter count of the conditional version (from 81M to 85.6M).
>
> As shown in the table below, compared to the original 81M version, the parameter-expanded unconditional model (85.6M) achieves comparable performance on ID testing and slightly improved results on OOD testing. However, it still largely underperforms the M2-based conditional variant in terms of OOD generalization, despite having an equivalent number of parameters. This supports our original claim that the observed performance gains stem from the proposed M2 modeling component, rather than simply model param capacity.
>
> | variant | ID-$\mu$ ($\downarrow$) | ID-$\sigma$ ($\downarrow$) |  ID-ME | OOD-$\mu$ ($\downarrow$) | OOD-$\sigma$ ($\downarrow$) | OOD-ME |
> |:--------:|:--------:|:-----------:|:------:|:---------:|:------------:|:------:|
> |   original w/o M2 (81M)  |   0.007   |     1.63    | 37.15 |    -14.66   |     54.95     | -12069.44 |
> |   expanded w/o M2 (85.6M)   |  -0.009   |  1.74      |  65.29   |   6.68   |    42.60     |    417.13   |
> |   ours (85.6M)   |   -0.02  |     0.71    |  32.09 |    0.51   |     2.32     |  -5.04 |
>
> >  I am a bit surprised that there is only averaging on the overlapping regions. I would assume that this would produce very blurry results and the fact that otherwise the images are processed independently would completely destroy the long-range correlation.
>
> **A2:** Thank you for raising this interesting point. We would like to further clarify the rationale behind this aggregation strategy from both empirical and physical perspectives below:
>
> - **Empirically**, we find that long-range coherence is influenced by the patching scheme, particularly the choice of step size during inference. In our reported experiments on the Taurus B213, we use a step size of k = 8 for 128×128 patches. This leads to substantial overlap between adjacent patches, which helps smooth transitions and mitigate visible artifacts. In contrast, using a larger step size (e.g., 32 or 64) with smaller overlap and often produces less natural aggregated maps with noticeable discontinuities.
>
> - **Physically**, the observational maps in our astro domain exhibit strong spatial correlations due to the underlying physics of molecular clouds. As a result, even though patches are processed independently during inference, they are not intrinsically independent in content, which is different from many standard image domains. This intrinsic correlation helps preserve global consistency when the patches are merged.
>
> > Any intuition why this memorization does not appear in your case but does in the case of Image models?
>
> **A3:** This is an insightful question. In the image generation, memorization is typically assessed via visual perceptual similarity between generated and training samples. However, such visual comparisons are not directly meaningful in the astro domain. In our case, although the predicted physical states (e.g., density maps) may appear visually similar at first glance when rendered in pixel space, these representations must be rigorously transformed back to their original physical domain (e.g., $10^{0}$ to $10^{7}$) under physical constraints for evaluation. This **nonlinear transformation significantly amplifies subtle differences**, so two maps that look similar visually actually represent very different states. As a result, “visual similarity” does not imply model memorization in physically meaningful sense. We believe this helps intuitively explain why we do not observe signs of memorization in our evaluations, even though similar generative architectures may exhibit this behavior in image domains.
>
> More broadly, as also discussed in response to several earlier questions, the application of ML frameworks to real-world astro problems introduces a number of non-trivial differences from vision tasks, which have often tended to be undervalued from a pure ML perspective. We are thus very grateful for the reviewer’s open attitude towards those differences.

---

> ### Author Response · Authors · 2025-08-04
>
> > We would like to take the opportunity to clarify that “paired DSB” is non-sensical as DSB and any Schrodinger Bridge and Optimal Transport (OT) based method must work on unpaired data as one of the goal of OT is to discover said pairing. We are aware of some confusion in the literature but we feel like this misunderstanding should be properly addressed.
>
> **A4:** We sincerely appreciate the reviewer’s commitment to conceptual rigor and are grateful for the opportunity to further reflect on this point. We fully agree that, from a classical SB/OT perspective, the problem is inherently unpaired, with its central purpose being to discover the coupling. That said, we believe this issue is *more than a sole matter of “right or wrong” terminology*. Below, we would like to offer some additional reflections.
>
> - **Terminological ambiguity in different contexts.**  While we understand the reviewer's concern, we feel that the use of terminology, such as “paired DSB”, is often grounded in specific background, training, and research priorities, given different starting points as research motivation. More specifically, while the ''paired nature" seems to contradict the **original foundation** of SD/OT problems, it does present a meaningful pragmatic role for distinguishing training constraints compared to its vanilla version from a **more ML technical** perspective (which is closer to our current use case).
>
>    An analogy could be drawn to a broader context of "diffusion", which has a deep history in physics (e.g., thermodynamic diffusion). Many core components of modern diffusion models in ML, such as discrete schedulers or fewer step distillations for large T2I models, deviate significantly from the continuous-time SDE or Fokker–Planck interpretations. Therefore, from a **physics-first perspective**, this could be considered a misuse; yet **from the ML side**, it reflects an evolving abstraction grounded more in practical algorithm design than physical fidelity. We believe these examples illustrate how the appropriateness of terminology often depends on disciplinary context and epistemological goals, rather than a single judgement of "right or wrong".
>
> - **What we could propose to do in this work.**  Although a consensus could be hard to reach within the community, we agree with the reviewer that it would be *helpful to clarify this in a more explicit way*. In particular, we will add a note explaining that while some prior ML literature adopts this terminology to distinguish between training configurations, the classical SB problem is fundamentally unpaired. We believe this clarification will help readers better navigate the distinction between traditional formulations and how they are instantiated in practical ML models.

---

### Official Review · Reviewer_Ey1R · 2025-06-19

**Clarity:** 2
**Significance:** 3
**Originality:** 2
**Rating:** 4
**Confidence:** 5

**Summary:**

This paper introduces Astro-DSB, a specialized Diffusion Schrödinger Bridge model for astrophysical inverse prediction tasks within Giant Molecular Clouds (GMCs). The authors adapt the I2SB framework with three modifications: (M0) pairwise marginal distribution matching, (M1) noise perturbation alignment with observational uncertainties, and (M2) observable enhancement throughout the diffusion trajectory. The method is evaluated on both synthetic magnetohydrodynamics (MHD) simulations and real observational data from Taurus B213, demonstrating improved performance over conventional astrostatistical methods and other ML approaches in both in-distribution (ID) and out-of-distribution (OOD) settings.

**Questions:**

1. How does the learned diffusion trajectory relate to the actual physical evolution of GMCs? The paper claims to capture "dynamic evolution processes" but doesn't establish this connection rigorously.

2.The choice of λ = 0.1 for the observational noise appears arbitrary. How sensitive are the results to this parameter, and how should it be chosen for different observational scenarios?

3. How does the method scale with the number of physical quantities to predict? Would the pairwise matching assumption (M0) become limiting for multi-output scenarios?

4.The comparison with conditional DDPMs uses the same training epochs (100 vs 400), but it's unclear if this represents convergence for both methods. Could the DDPM performance improve with longer training?

5.What specific aspects of the DSB formulation enable better OOD generalization compared to discriminative approaches? The explanation remains largely empirical.

**Ethical Concerns:**

["NO or VERY MINOR ethics concerns only"]

**Final Justification:**

as I said, pros: The authors cleared up my questions about the physical dynamics connection (Q1) and noise sensitivity (Q2). Reviewer 8Wk8 made a good point about M1/M2, the ablations show it's not just more parameters but actual modeling improvements. The ENZO vs ORION2 OOD setup is actually stronger than I first thought since they use different physics, not just initial conditions. cons: Still think the novelty over I2SB is limited, but the 75% faster training and working on real data makes it worthwhile for the astro community? So I cant give a score more than 4.

**Limitations:**

The approach is limited to paired training data, which may not always be available in real observational scenarios. please also refer to the cons and questions. thank you.

**Paper Formatting Concerns:**

Mathematical notation lacks consistency throughout (especially in equations 3-6).

**Quality:**

3

**Strengths And Weaknesses:**

pros
1.The application of diffusion bridge models to astrophysical inverse problems represents a meaningful extension beyond traditional computer vision tasks, addressing scientifically important problems in star formation research.

2. Unlike heuristic OOD definitions in vision tasks, this work provides principled OOD evaluation through variations in initial physical conditions and dominant physical processes, making the generalization assessment more scientifically grounded.

3. The evaluation on real observational data from Taurus B213 demonstrates practical applicability beyond synthetic simulations, which is crucial for scientific applications.

4.The reported 75% reduction in training time compared to conditional DDPMs while achieving superior performance is practically significant.

con s:
1. The core technical contributions are relatively incremental modifications to existing I2SB framework. The three proposed modifications (M0, M1, M2) are straightforward adaptations rather than fundamental algorithmic innovations.

2.The paper lacks theoretical analysis of why DSB should be particularly well-suited for astrophysical inverse problems compared to other generative approaches. The connection between the physical dynamics of GMCs and the SDE formulation is not rigorously established.

3.The synthetic data comes from only two MHD codes (ENZO and ORION2), potentially limiting the generalizability of conclusions about OOD performance.

---

> ### Author Rebuttal · Authors · 2025-07-30
>
> We thank the reviewer for the valuable feedback and are encouraged by the recognition of our principled OOD setup, evaluation on real observations, and the practical significance of improved training efficiency. Below we address the raised questions in detail.
>
> ---
> **Q1: How does the learned diffusion trajectory relate to the actual physical evolution of GMCs?**
>
> **A1:** We appreciate the insightful question. Our interpretation of dynamics is established through explicit state alignment in a data-driven manner, which we will further elaborate on from the following perspectives:
>
> - **The nature of observational inversion problems.** As noted in L95–101 and L115–125, while the physical forward process $\mathcal{H}$ from initial condition $x_0$ to current physical state $x_1$ is governed by known equations and simulated explicitly, the inverse prediction from observables $y$ is **fundamentally intractable**, similar to many other scientific inverse problems. As a result, even established astrophysical methods like the DCF approach (Davis 1951; Chandrasekhar & Fermi 1953) are fundamentally statistical rather than derived from physical laws. This ill-posed nature makes data-driven machine learning approaches particularly suitable for such tasks.
>
> - **How can our method better capture underlying dynamics in this case?** Despite the intractability, it is well accepted in astrophysics that *the observables $y$ are indirectly shaped by the underlying dynamics of GMC evolution*. Our assumption is that, by aligning observable $y$ and physical state $x_1$ via a data-driven bridge, we can implicitly capture this evolution. The key hypothesis is: if our model can consistently align the distributions of physical states across varying observations, we gain greater confidence that the learned transitions reflect meaningful astrophysical dynamics, **given these states are ultimately also governed by physical laws (despite they remain to be unknown yet in astro)**. Compared to SOTA cDDPMs, which rely on an artificial Gaussian prior (see L150–158), our bridge formulation avoids this assumption, resulting in more physically grounded state alignment and faster convergence.
>
> That said, We acknowledge that directly encoding physical equations into the generative process is a longer-term goal, but believe our method offers a principled and empirically supported step **over existing ML and astrophysical approaches**.
>
> ---
> **Q2: The choice of λ = 0.1 for the observational noise appears arbitrary. How sensitive are the results to it?**
>
> **A2:** Thank you for the comment. The choice of $\lambda = 0.1$ is not arbitrary, but follows statistical conventions as we mentioned in L.180.
>
> In **existing astro literature**, noise is commonly introduced both by the instrumentation (the telescope) and by foreground/background gas emission with a signal-to-noise ratio (SNR) at the order of 10.  For instance, the noise level from the column density map from the Five College Radio Astronomical Observatory (FCRAO) based on the 12CO observations [1] is around 0.28K, v.s. signal of 1-7K. Similarly, the extinction maps derived from Spitzer infrared observations toward several molecular clouds [2,3] shows ~10% uncertainties.
>
> In the **ML context**, we additionally varied $\lambda$ from 0.1 to 0.3. Overall, ID performance remains stable, while a larger $\lambda$ slightly degraded OOD results but **still outperformed** DDPMs.
>
> | $\lambda$ | ID-$\mu$ ($\downarrow$) | ID-$\sigma$ ($\downarrow$) |  ID-ME | OOD-$\mu$ ($\downarrow$) | OOD-$\sigma$ ($\downarrow$) | OOD-ME |
> |:--------:|:--------:|:-----------:|:------:|:---------:|:------------:|:------:|
> |    0.1   |   -0.02  |     0.73    |  32.09 |    0.51   |     2.32     |  -5.04 |
> |   0.15   |   0.03   |     1.61    | 145.04 |    2.52   |     4.91     | 313.00 |
> |    0.2   |   -0.02  |     1.58    | 131.34 |    2.16   |     5.33     | 186.37 |
> |   0.25   |   -0.02  |     1.41    |  90.65 |    2.19   |     5.28     | 164.43 |
> |    0.3   |   0.03   |     1.74    | 141.40 |    2.71   |     5.71     | 240.55 |
>
> ---
> **Q3: How does the method scale with the number of physical quantities to predict? Would the pairwise matching assumption (M0) limit multi-output scenarios?**
>
> **A3:** **Physically**, M0 stems from the intrinsic nature of observational inverse prediction tasks. Specifically, under a fixed observational process $\mathcal{H}$, the physical state $x_1$ should **produce a deterministic observation $y$**. Therefore, multi-output scenarios (i.e., multiple $x_1$ for a given $y$) should already be **encoded in the simulation data** rather than arising from the formulation ambiguity.
>
> As for **ML scalability**, DSB actually scales better than SOTA DDPMs, which require modeling across three distributions (i.e., $y$, $x_1$, and a Gaussian prior) and introduce additional complexity and potential misalignment. In contrast, DSB avoids the artificial Gaussian prior by directly modeling the OT bridge between $y$ and $x_1$, simplifying learning and improving generalization. Similar findings have been reported in both interdisciplinary domains [4] and image generation literature [5].
>
> **Additional empirical support:** In our experiments, predicting magnetic fields (a vector field) from observations shows scalability across output dimensionality beyond scalar density. The successful results on both tasks empirically demonstrate the scalability of our method w.r.t. the dimensionality and complexity of physical quantities.
>
> ---
> **Q4: The comparison with cDDPMs uses the same training epochs, but unclear if this represents convergence for both methods. Could the DDPM improve with longer training?**
>
> **A4:** We ensured that the reported training epochs correspond to convergence for each method. In Appendix C.2, Fig. 6, we also showed different training curves, and the performance of DDPMs no longer improves with longer training.
>
> ---
> **Q5: What specific aspects of the DSB enable better OOD generalization compared to discriminative approaches?**
>
> **A5:** Thank you for the insightful question. The key difference in DSB lies in its distribution-level modeling between the observable and the physical state, as opposed to directly learning a discriminative point-to-point mapping. We hypothesize that this design improves OOD generalization through the following:
>
> - **Data distribution**: bridge-based generative modeling mitigates distributional mismatch. Discriminative models often struggle with OOD generalization due to substantial covariate shifts between input and output spaces. In contrast, DSB explicitly learns the stochastic evolution from $y$ to $x_1$ via a bridge, offering greater flexibility and *broader coverage over the underlying data distribution*. Under a shifted data sample, the model is more likely to operate within a plausible region of the learned data manifold, resulting in more reliable predictions.
>
> - **Physical modeling**: DSB enables explicit alignment between physically meaningful states, which encourages implicit consistency with the natural dynamics of GMC evolution. As discussed in A1, both boundary states in astro are governed by hidden physical processes, and the intermediate transitions are often treated statistically (e.g., Gaussian processes). The DSB, grounded in optimal transport, naturally captures such transitions and implicitly respects these physical assumptions.
>
> ---
> **Limited technical nolvety over I2SB.**
>
> **A6:** We understand the reviewer’s concern, however, we would like to respectfully clarify that this work is **not merely a methodology** paper, and the vanilla I2SB framework **is not directly applicable** in these inverse tasks.
>
> First, our proposed method demonstrates **consistent improvements in both prediction accuracy and training efficiency** compared to both conventional astrostatistical approaches and recent ML SOTA baselines, as stated in our abstract.
>
> Second, we emphasize the generalization ability under **physically motivated out-of-distribution (OOD) settings and validate its robustness on real astronomical observations (Taurus B213)**, which are rarely addressed in standard ML contexts. This constitutes a non-trivial and essential step toward applying machine learning methods in scientific domains where interpretability, robustness, and physical grounding are crucial.
>
> Finally, the vanilla I2SB framework is not **directly applicable to inverse prediction problems**. Originally designed for image translation, it corresponds to our setting without modules M1 and M2. As shown in our ablation studies, both components are essential for achieving strong ID and OOD performance.
>
> In this sense, our work contributes not only methodologically, but also meaningfully bridges machine learning and astrophysical research through practical and verifiable advancements.
>
> ---
> **Limited OOD generalization given two MHD codes (ENZO and ORION2).**
>
> **A7:** We believe that our experiments provided strong OOD generalization ability. As noted in L218–223, beyond differing initial physical conditions, ENZO and ORION2 have also **adopt different physical assumptions**: ENZO includes self-gravity, magnetic fields, radiative heating and cooling, while ORION2 excludes gravity but includes driven turbulence. These differences lead to distinct simulation behaviors and observables to ensure the robustness of OOD generalization. Additionally, our test on real observable data from Taurus B213 also supports the generalization claim.
>
> ---
> [1] Large-Scale Structure of the Molecular Gas in Taurus Revealed by High Linear Dynamic Range Spectral Line Mapping, APJ 2008
>
> [2] Spectroscopic Infrared Extinction Mapping as a Probe of Grain Growth in IRDCs, APJ 2015
>
> [3] Denoising Diffusion Probabilistic Models to Predict the Density of Molecular Clouds, APJ 2023
>
> [4] Physics-aligned field reconstruction with diffusion bridge, ICLR 2025
>
> [5] Simple diffusion: End-to-end diffusion for high resolution image, ICML 2023

---

### Official Review · Reviewer_8Wk8 · 2025-07-02

**Clarity:** 2
**Significance:** 1
**Originality:** 1
**Rating:** 4
**Confidence:** 4

**Summary:**

This paper applies Diffusion Schrödinger Bridge (DSB) models to astrophysical systems, focusing on Giant Molecular Clouds (GMC).

**Questions:**

- Could the authors clarify how the cDDPMs were implemented? Was the prior simply replaced with a Gaussian, or were other modifications made?

- In Equation (5), it seems there is a typo: the RHS should likely use $x_1$ rather than $x_t$.

- In Equation (6), the loss term should be written as a squared norm.

**Ethical Concerns:**

["NO or VERY MINOR ethics concerns only"]

**Final Justification:**

In the initial review phase, I felt that the overall motivation and methodology closely followed I2SB, and the experimental results appeared unconvincing, particularly since the proposed method showed performance comparable to cDDPM in Tables 1 and 2, and the ablation studies in Table 3 did not consistently outperform baseline metrics. However, I now better appreciate the novelty of this work, especially the authors’ clarification regarding the significance of the OOD generalization results. The emphasis on out-of-distribution performance has helped clarify the contribution. I now recognize that the proposed modifications in M1 and M2 play a meaningful role in balancing fidelity and generalization, which I believe strengthens the paper’s overall contribution.

**Limitations:**

Please refer to the points above on the lack of methodological novelty and limited depth of analysis.

**Paper Formatting Concerns:**

I do not see any formatting issues.

**Quality:**

2

**Strengths And Weaknesses:**

- The methodology is technically reasonable and follows established frameworks for DSB-based modeling.

- However, the contribution appears quite incremental. The training procedure closely mirrors existing approaches such as I2SB and Bridge-TTS, and the inference relies on a standard aggregation technique. I do not see a clear methodological innovation that sets this work apart from prior DSB literature.

- Additionally, I believe the analysis is weak. While the results include robustness and generalization of the proposed method, there is no discussions or insights into why the proposed method performs better than baselines, or what aspects of the astrophysical context might benefit specifically from this approach. Additionally, I don't see the clear improvements compared to cDDPM,

- The explanation of the comparisons are missing, both in the main text and appendix.  Especially, I do not see how cDDPM is implemented.

---

> ### Author Rebuttal · Authors · 2025-07-30
>
> We appreciate the comments from the reviewer, below we provide our detailed responses to raised questions.
>
> ---
> **No discussions or insights into why the proposed method performs better than baselines, or what aspects of the astrophysical context might benefit specifically from this approach. Additionally, I don't see the clear improvements compared to cDDPM.**
>
> **A1:** All the requested discussions have been **comprehensively included in the original submission**.
>
> - **A.1.1:** Insights into why the proposed method performs better than baselines.
>
>    - **For astrostatistical baselines, L95-101:** “*Traditionally in astronomy, such observational inverse prediction problems have been commonly approached via astrostatistical methods, which often combine theoretical simplifications with statistical priors…*” In other words, due to the intractable nature of observational inverse prediction tasks, conventional astro methods, such as the DCF approach (Davis 1951; Chandrasekhar & Fermi 1953), are fundamentally statistical rather than governed by explicit physical laws. Our proposed data-driven ML approach offers advantages by learning complex nonlinear mappings without relying on oversimplified assumptions.
>
>    - **For existing ML baselines, particularly recent cDDPMs, L150-158:** “*... conditional DDPMs introduce unnecessary Gaussian prior assumptions, which are not naturally aligned with the physics of GMCs. Our proposed Schrödinger bridge framework removes this assumption by directly modeling the transition between observable $y$ and the true physical distribution $x_1$, resulting in a distribution-level probabilistic mapping without reliance on artificial prior structures, thus offering better ML interpretability compared to conditional DDPMs.*”
>
> - **A.1.2:** Aspects of the astrophysical context might benefit specifically from this approach.
>
>    - As described in L150–158, the proposed formulation enables state-to-state alignment between the observed data and the underlying physical states, which provides an effective inductive bias for learning meaningful dynamics. Moreover, given the intractable and statistical nature of inverse prediction tasks, the advantage of our method lies in bypassing the need for artificial priors (like Gaussian noise) and instead leveraging observable-informed transitions to achieve better dynamic fidelity and generalization.
>
> - **A.1.3:** Improvements compared to cDDPM.
>
>    - **Performance-wise, Tab.1, Tab.2 and Fig. 3, L291-297**: “*...particularly demonstrating robust superiority in OOD predictions, as indicated by the lower mean (µ) and standard deviation (σ) of the relative error distribution…*”. For instance, in Table 2, our ID density prediction achieves a mean error $\mu$ of -0.02, compared to -0.05 from cDDPM. In the OOD case, the relative error is 0.51 for Astro-DSB, compared to 2.88 for cDDPMs.
>
>    - **Efficiency-wise, L297-298, Appendix C.2:** “*In particular, the training cost of Astro-DSB is only 25% of the previous SOTA conditional DDPMs method.*” Figure 6 in Appendix C.2 further shows the faster convergence of Astro-DSB during training.
>
> ---
> **Could the authors clarify how the cDDPMs were implemented? Was the prior simply replaced with a Gaussian, or were other modifications made?**
>
> **A2:** The cDDPM baselines are introduced in prior literature [1, 2], and we followed the publicly available codebases for reproduction in our paper. This includes the standard conditional DDPM framework with 1000 diffusion steps, where the observables $y$ are passed as conditioning inputs $c$ into the U-Net at each step during training to model $p(x_{t-1}|x_t, c)$. No explicit aggregate sampling techniques were used.
>
> At a high level, one core difference between cDDPMs and DSB formulation lies in the starting distribution: cDDPMs rely on a fixed Gaussian prior, whereas DSB directly models a bridge between the observation $y$ and the target distribution $x_1$. As explained in L150–158, this difference enables better state alignment and contributes to significantly faster convergence (25% of the cDDPM training), as the distributional distance between the two endpoints is smaller and more physically meaningful.
>
> In summary, the cDDPM implementations are directly based on prior literature, but we would be happy to clarify this further or include additional implementation details in the revised version if the reviewer finds it helpful.
>
> ---
> **In Eq. (6), it seems there is a typo: the RHS should likely use $x_1$  rather than $x_t$. In Eq. (7),  the loss term should be written as a squared norm.**
>
> **A3:** We have carefully reviewed Eq. (6) and Eq. (7) and confirm that there are no typos. In Eq. (6), our intention is to present the posterior distribution of the intermediate state $x_t$ conditioned on $x_1$, which is consistent with the formulation. As for Eq. (7), we use the L1 norm rather than the squared L2 norm, and this has been a rather standard operation as in prior literature [3].
>
> ---
> **Contribution appears quite incremental.**
>
> **A4:** We understand the reviewer’s concern, however, we would like to respectfully clarify that this work is not merely a methodology paper, and the vanilla I2SB framework is not directly applicable in these inverse tasks.
>
> First, our proposed method demonstrates **consistent improvements in both prediction accuracy and training efficiency** compared to both conventional astrostatistical approaches and recent ML baselines. As stated in the abstract: “*From the astrophysical perspective, our proposed paired DSB method improves interpretability, learning efficiency, and prediction performance over conventional astrostatistical and other machine learning methods.*”
>
> Second, we emphasize the generalization ability of the proposed method under **physically motivated out-of-distribution (OOD) settings and validate its robustness on real astronomical observations (Taurus B213)**, which are rarely addressed in standard ML contexts. This constitutes a non-trivial and essential step toward applying machine learning methods in scientific domains where interpretability, robustness, and physical grounding are crucial.
>
> Finally, the vanilla I2SB framework is not **directly applicable to inverse prediction problems**. Originally designed for image translation, it corresponds to our setting without modules M1 and M2. As shown in our ablation studies, both components are essential for achieving strong ID and OOD performance.
>
> In this sense, our work contributes not only methodologically, but also meaningfully bridges machine learning and astrophysical research through practical and verifiable advancements.
>
> ---
> [1] Denoising Diffusion Probabilistic Models to Predict the Density of Molecular Clouds, APJ 2023
>
> [2] Exploring Magnetic Fields in Molecular Clouds through Denoising Diffusion Probabilistic Models, APJ 2025
>
> [3] Physics-aligned field reconstruction with diffusion bridge, ICLR 2025

---

> > ### Comment · Reviewer_8Wk8 · 2025-08-04
> >
> > Thank you for the detailed response. I have carefully reviewed both the manuscript and the authors’ rebuttal.
> >
> > In the initial review phase, I felt that the overall motivation and methodology closely followed I2SB, and the experimental results appeared unconvincing, particularly since the proposed method showed performance comparable to cDDPM in Tables 1 and 2, and the ablation studies in Table 3 did not consistently outperform baseline metrics. However, I now better appreciate the novelty of this work, especially the authors’ clarification regarding the significance of the OOD generalization results. The emphasis on out-of-distribution performance has helped clarify the contribution. I now recognize that the proposed modifications in M1 and M2 play a meaningful role in balancing fidelity and generalization, which I believe strengthens the paper’s overall contribution.
> >
> > That said, I have several comments on the method as a whole, and would appreciate the authors’ response to the following points:
> >
> > - If M1 and M2 are indeed critical to the method’s success and represent a point of departure from I2SB, I believe it would be helpful to emphasize this distinction more clearly. In particular, the conditioning on $y$ seems essential (as discussed in the paper), and this formulation is conceptually different from I2SB. It may be more appropriately interpreted as learning a conditional SB (or a conditional Doob’s $h$-transform), where the process bridges $y* \mathcal{N}(0, \lambda I)$ to $x_1$, conditioned on the observation $y$. This conceptual framing could clarify the theoretical positioning of the work and differentiate it more clearly from prior methods.
> >
> > - In the authors response A.3, the use of L1 loss deviates from the standard SB framework, which is theoretically grounded in the use of an L2 loss. While I understand this may be a practical design choice, it would be helpful to include an ablation comparing L1 and L2 losses, to empirically justify this deviation.
> >
> >
> > I would happy to increase the score to 4 after the response.

---

> > > ### Author Response · Authors · 2025-08-05
> > >
> > > We very much appreciate the reviewer’s careful reading of our rebuttal responses, thoughtful engagement, and valuable follow-up comments, and are glad to further address the points raised below.
> > >
> > > > On the conceptual framing clarifications: If M1 and M2 are indeed critical to the method’s success and represent a point of departure from I2SB, I believe it would be helpful to emphasize this distinction more clearly.
> > >
> > > **A1:** We thank the reviewer for the constructive feedback. While our original manuscript puts a greater emphasis on the methodological differences from previously used frameworks (e.g., cDDPMs) in astrophysical observational prediction tasks for *the Astro field*, we agree that making the conceptual distinctions for M1 and M2 more explicit would significantly help clarify our contribution. In particular, the reviewer’s suggestion to interpret M2 as a conditional SB or a conditional Doob’s $h$-transform is a valuable perspective, especially for researchers *from the broader ML community*. We will incorporate this clarification into the revised version to better highlight the modelling distinctions.
> > >
> > > In addition to the ablation studies already presented, we have also conducted further controlled experiments to rigorously assess the impact of M2 while matching model capacity (as suggested by R-fbj1). Specifically, we compare (i) an expanded U-Net with equivalent parameters and (ii) a conditional setup with zero-padded $y$, and report the results below. These results confirm our original claim that the observed improvements stem from the M2 modeling strategy, rather than increased parameter scale.
> > >
> > > | variant | ID-$\mu$ ($\downarrow$) | ID-$\sigma$ ($\downarrow$) |  ID-ME | OOD-$\mu$ ($\downarrow$) | OOD-$\sigma$ ($\downarrow$) | OOD-ME |
> > > |:--------:|:--------:|:-----------:|:------:|:---------:|:------------:|:------:|
> > > |   expanded w/o M2 (85.6M)   |  -0.009   |  1.74      |  65.29   |   6.68   |    42.60     |    417.13   |
> > > |   void cond (85.6M)   |   -0.10   |     2.12    | 62.72 |    6.16   |     12.23     | 381.70 |
> > > |   ours (85.6M)   |   -0.02  |     0.71    |  32.09 |    0.51   |     2.32     |  -5.04 |
> > >
> > >
> > > > It would be helpful to include an ablation comparing L1 and L2 losses, to empirically justify this deviation.
> > >
> > > **A2:** We appreciate the reviewer’s suggestion and have conducted an additional ablation comparing the use of L1 and L2 losses. The results are reported below. We find that both loss functions yield comparable performance in both ID and OOD settings, with L2 even exhibiting slightly better generalization in the OOD case, which suggests that *the core effectiveness of our method remains stable across reasonable loss formulations*. We are glad to integrate these additional results and discussions into the revised paper.
> > >
> > > | loss| ID-$\mu$ ($\downarrow$) | ID-$\sigma$ ($\downarrow$) |  ID-ME | OOD-$\mu$ ($\downarrow$) | OOD-$\sigma$ ($\downarrow$) | OOD-ME |
> > > |:--------:|:--------:|:-----------:|:------:|:---------:|:------------:|:------:|
> > > |   L1   |   -0.02  |     0.71    |  32.09 |    0.51   |     2.32    |  -5.04 |
> > > |   L2  |   0.03   |     0.86    | 47.26 |    0.47   |     2.11     | 6.78 |

---

> > > > ### Comment · Reviewer_8Wk8 · 2025-08-08
> > > >
> > > > I appreciate author for further clarification. I raised score to 4.

---

### Official Review · Reviewer_RTRd · 2025-07-02

**Clarity:** 3
**Significance:** 2
**Originality:** 2
**Rating:** 4
**Confidence:** 3

**Summary:**

This paper proposes a new method called Astro-DSB with a pairwise domain assumption for solving the problem of observational inverse prediction in giant molecular clouds related to star formation. Through physical simulations and real Taurus B213 observations, it shows the method enhances interpretability, learning efficiency, and prediction performance over traditional astrostatistical and machine learning approaches.

**Questions:**

1. Could analytic or learned physical drifts be inserted into the bridge to improve interpretability and extrapolation?
2. What exactly delivers the four-fold training-time speed-up over the cDDPM baseline (shorter schedule, faster convergence, architectural changes)?
3. Do multiple stochastic samples yield meaningful uncertainty estimates for astronomers, or is the output effectively deterministic?

**Ethical Concerns:**

["NO or VERY MINOR ethics concerns only"]

**Final Justification:**

Overall, I appreciate the authors’ careful and detailed engagement with the previous comments. The authors thoughtfully position their work not as a final solution, but as a meaningful step forward in building machine learning models with stronger physical alignment for astrophysical inverse problems, acknowledging the real challenges of encoding fundamental physics. The justification for the OOD evaluation is strong by emphasizing different physical assumptions and the test scenarios are not arbitrary, but scientifically relevant. Their choice of a data-driven approach is well-argued, especially given the absence of analytical solutions in this domain, and it reflects a principled approach to tackling complex inverse problems. Regarding uncertainty, since the outputs are deterministic, they’ve validated consistent results across multiple runs, which adds credibility.

**Limitations:**

1. The dynamics are purely data-driven. Conservation laws and intermediate physical states are not enforced.
2. Iterative sampling can be slow for large-scale deployment without further acceleration or distillation.

**Paper Formatting Concerns:**

No formatting issues.

**Quality:**

3

**Strengths And Weaknesses:**

Strengths:

Originality: It innovatively adapts DSB models to astrophysical dynamics, introducing the Astro-DSB framework with pairwise domain assumptions—a novel extension beyond traditional visual synthesis applications. It also introduces noise-aligned conditioning and step-wise observation injection.

Quality: The research is rigorously supported by comprehensive experiments on both simulated magnetohydrodynamic data and real Taurus B213 observations. Quantitative results highlight Astro-DSB’s superior performance (e.g., reduced relative error μ=0.51±2E-4 in OOD density prediction) and 75% faster training than conditional DDPMs, validated through ablation studies and comparative analyses against astrostatistical/ML baselines.

Clarity: The paper is systematically structured, clearly outlining the problem setup, technical designs and evaluation metrics. Pictures and tables effectively communicate results.

Significance: The work shows that learning a full distributional trajectory can deliver stronger OOD robustness and lower training cost than conventional conditional diffusion or direct regression, with clear impact for scientific machine learning.

Weaknesses:

1. Incomplete physical modeling: The Astro-DSB framework does not explicitly encode the full forward physical equations governing GMC evolution. This limits its interpretability as a physics-driven model, as the learned dynamics may not fully align with underlying astrophysical laws.
2. Limited OOD scenario validation: While OOD testing uses different MHD codes (ENZO vs. ORION2), the study focuses on specific initial conditions. It lacks validation across broader physical parameter spaces.

---

> ### Author Rebuttal · Authors · 2025-07-30
>
> We appreciate the valuable feedback from the reviewer. We’re also grateful for the reviewer’s recognition of the paper’s strengths, including the novel adaptation of DSBs to astrophysical dynamics, the strong OOD performance, and the significantly faster training compared to conditional DDPMs, highlighting both the methodological contribution and practical impact in scientific ML settings.
>
>
> Below, we provide our detailed responses to the raised concerns and questions, particularly emphasizing how our proposed method contributes to *“better physical alignment over existing literature in astrophysics and other ML methods” under the intractable nature of observational inverse prediction problems*.
>
>
> ---
> **W: Incomplete physical modeling as the learned dynamics may not fully align with underlying astrophysical laws.**
>
>
> **A1:** While we understand the reviewer’s concern, we would like to respectfully clarify that fully encoding the complete set of physical laws governing GMC evolution remains inherently difficult. It is a broader goal that requires sustained, collaborative efforts from the entire community and is unlikely to be fully addressed by a single paper.
>
>
> We posit ourselves as a step forward with a better alignment ML formulation **compared to existing available ML and conventional astro methods** tackling this line of problems (see our A3 below for details), rather than a framework that fully addresses this long-term challenge. As we also noted in the Limitations section (L332–338), this difficulty stems largely from the **ill-posed and intractable nature** of inverse prediction tasks in astrophysics. In fact, even for conventional astrophysical approaches, such as the DCF baseline (Davis 1951; Chandrasekhar & Fermi 1953) included in our comparison, are **statistical methods rather than exact physical solvers**, which often rely on simplified physical assumptions (L95-101).
>
>
> ---
> **W: OOD testing using different MHD codes only focuses on specific initial conditions.**
>
>
> **A2:** **Our OOD scenarios differ in more than just initial conditions.** As explained in L218–223, ENZO (ID) and ORION2 (OOD) adopt different physical assumptions: ENZO includes self-gravity, magnetic fields, radiative heating and cooling, while ORION2 excludes gravity but includes driven turbulence. These differences lead to distinct simulation behaviors and observables, which also motivate us to use them as OOD test cases. Additionally, we believe that the real observational data also serve as strong OOD testing cases.
>
>
> ---
> **Q: Could analytic or learned physical drifts be inserted into the bridge to improve interpretability and extrapolation?**
>
>
> **A3:** We appreciate the insightful question. For observational inverse prediction problems, there are no known analytical physical descriptions, as these tasks **are inherently intractable**. This is also why conventional astrophysical methods we compared, such as the DCF approach (Davis 1951; Chandrasekhar & Fermi 1953), **are fundamentally statistical** rather than governed by explicit physical laws. While the forward physical evolution $\mathcal{H}$ can be simulated from initial conditions, there are no explicit physical constraints defining the mapping $\mathcal{F}$ between the true physical states and the observables (as discussed in L115–125).
>
>
> This ill-posed *physical nature makes data-driven machine learning approaches particularly suitable* for tackling such astro problems. This is also the intuition behind our approach: by aligning ML and physical states through a data-driven bridge, we aim to implicitly capture the underlying dynamics. The ultimate hypothesis is that if the model can consistently align distributions of physical states across varying observations, it increases our confidence that the learned process reflects meaningful astrophysical evolution, since these states are ultimately governed by physical laws (equations). But again, this remains a long-term research goal. That said, we believe our work represents a meaningful step forward compared to existing approaches in both astrophysics and machine learning (see also A4 for further discussion).
>
>
> ---
> **Q: What exactly delivers the four-fold training-time speed-up over the cDDPM baseline?**
>
>
> **A4:** The speed-up primarily stems from **the reduced distributional gap in our formulation**. In cDDPM, the model learns to map from a standard Gaussian to the physical state $x_1$, which involves a large distributional distance. In contrast, Astro-DSB learns a bridge between the observation $y$ and the target $x_1$, which are more closely aligned. This smaller gap leads to more efficient training and faster convergence. Similar trends with comparison to DDPMs have also been reported in other works such as [1].
>
>
> This further demonstrates the advantage of our formulation in aligning with physical states, since the Gaussian prior in cDDPM lacks physical meaning (as explained in L150–158, "...conditional DDPMs introduce unnecessary Gaussian prior assumptions..."), whereas our method is closer to a real astrophysical observational scenario.
>
>
> ---
> **Q: Do multiple stochastic samples yield meaningful uncertainty estimates for astronomers, or is the output effectively deterministic?**
>
>
> **A5:** Thank you for the question. In the physical context, particularly for the simulation data, the observable $y$ is **theoretically deterministic** given the physical state $x_1$ under a pre-defined observational process. Therefore, while uncertainty estimation is indeed an important research direction in many scientific modeling, this specific task setup *does not explicitly require it*.
>
> From the ML perspective, we conducted several trials of stochastic sampling using different random seeds and observed minimal variation across runs (on the order of $10^{-3}$ to $10^{-4}$, negligible in astro studies), as reported in our tables. This is consistent with prior literatures that use cDDPMs as the framework [2,3].
>
> ---
> **Iterative sampling can be slow for large-scale deployment without further acceleration or distillation.**
>
>
> **A6:** Thank you for the comment. In our original experiments, we also tested the DDIM sampling strategy with skipped steps during inference, as mentioned in L287–289. We observed that this acceleration **did not lead to significant performance degradation** in our scientific prediction tasks. This suggests that faster sampling is *indeed feasible in practice*. That said, we agree that further exploration of advanced acceleration techniques, such as distillation or adaptive schedulers, remains a valuable direction for improving deployment efficiency in large-scale scientific applications.
>
> ---
> [1] Physics-aligned field reconstruction with diffusion bridge, ICLR 2025
>
>
> [2] Denoising Diffusion Probabilistic Models to Predict the Density of Molecular Clouds, APJ 2023
>
>
> [3] Exploring Magnetic Fields in Molecular Clouds through Denoising Diffusion Probabilistic Models, APJ 2025

---

> > ### Comment · Area_Chair_eexx · 2025-08-05
> >
> > Dear reviewer RTRd,
> >
> > After considering the rebuttal to your review and the other reviews/rebuttals, how and why has this affected your position on this submission? Please reply with an official comment (not just the mandatory acknowledgement) reflecting your current view, any follow-up questions/comments etc.
> >
> > Note the Aug 6 AoE deadline, make sure to respond in time for the authors to be able to submit a response if necessary.

---

### Note · Authors · 2025-08-12

As our final remarks, we would like to thank the reviewers for their constructive engagement during the rebuttal phase, which has helped reach a clearer shared understanding of our work’s contributions. In particular, we are encouraged that the initial concern regarding the technical novelty of our method was addressed through additional clarifications and targeted ablations, which highlighted the distinct roles of M1 and M2 in improving OOD generalization beyond prior I2SB-based approaches for image translations.

Beyond methodological novelty, this work also establishes a new ML baseline and evaluation protocol for real astrophysical inverse problems, helping bridge the gap between ML models and domain-specific physics. In particular, the demonstrated robustness on both synthetic simulations and real observational data from the Taurus B213 region underscores the practical relevance of our approach. We believe this cross-disciplinary contribution, with the promising ID and OOD performance, will be of interest to both the ML and astrophysics communities.

We once again thank the reviewers and AC for their time, thoughtful feedback, and constructive discussions.

---

### Decision · Program_Chairs · 2025-09-17

**Decision:**

Accept (poster)

**Comment:**

The authors present Astro-DSB and apply it to a variety of applications in astrophysics. The method primarily builds on diffusion Schrödinger bridge approaches in the literature and with tailored, but fairly straightforward, modifications apply these to astrophysics. The method, Astro-DSB, is shown to perform better than or comparable to standard diffusion baselines with a significant improvement in training time.

The discussion focused on the concerns about methodological novelty, the significance of the experimental contributions, partially alleviated by the rebuttal, as well as the relevance for a machine learning conference. Ultimately, the discussion concluded that this paper is more of an application paper where the experimental validation and insights are potentially useful as a baseline and evaluation protocol for astrophysics in ML. The reviewers reached a consensus recommending weak accept for the potential utility for the astrophysics and AI4Science communities, which I don't see a strong reason to disagree with.